# The REST remodeling complex protects genomic integrity during embryonic neurogenesis

Tamilla Nechiporuk[1], James McGann[1], Karin Mullendorff[1], Jenny Hsieh[2,3], Wolfgang Wurst[4,5,6,7], Thomas Floss[4], Gail Mandel[1]*

[1]Vollum Institute, Howard Hughes Medical Institute, Oregon Health and Science University, Portland, United States; [2]Department of Molecular Biology, University of Texas Southwestern Medical Center, Dallas, United States; [3]Hamon Center for Regenerative Science and Medicine, University of Texas Southwestern Medical Center, Dallas, United States; [4]Institute of Developmental Genetics, Helmholtz Zentrum München, Neuherberg, Germany; [5]Technische Universität München, Munich, Germany; [6]German Center for Neurodegenerative Diseases (DZNE), Munich, Germany; [7]Munich Cluster for Systems Neurology (SyNergy), Ludwig-Maximilians-Universität, Munich, Germany

**Abstract** The timely transition from neural progenitor to post-mitotic neuron requires down-regulation and loss of the neuronal transcriptional repressor, REST. Here, we have used mice containing a gene trap in the *Rest* gene, eliminating transcription from all coding exons, to remove REST prematurely from neural progenitors. We find that catastrophic DNA damage occurs during S-phase of the cell cycle, with long-term consequences including abnormal chromosome separation, apoptosis, and smaller brains. Persistent effects are evident by latent appearance of proneural glioblastoma in adult mice deleted additionally for the tumor suppressor p53 protein (p53). A previous line of mice deleted for REST in progenitors by conventional gene targeting does not exhibit these phenotypes, likely due to a remaining C-terminal peptide that still binds chromatin and recruits co-repressors. Our results suggest that REST-mediated chromatin remodeling is required in neural progenitors for proper S-phase dynamics, as part of its well-established role in repressing neuronal genes until terminal differentiation.

*For correspondence: mandelg@ohsu.edu

**Competing interests:** The authors declare that no competing interests exist

## Introduction

The transcriptional repressor REST (also called NRSF; *Schoenherr and Anderson, 1995*) binds to thousands of coding and non-coding genes that, as an ensemble, are required for the terminally differentiated neuronal phenotype (*Bruce et al., 2004*; *Conaco et al., 2006*; *Otto et al., 2007*; *Mortazavi et al., 2006*). In situ hybridization analysis shows a striking reciprocal pattern of REST expression to genes expressed within the developing nervous system, consistent with its role as a repressor (*Schoenherr and Anderson, 1995*; *Chong et al., 1995)*. In differentiating cells in culture, REST is down-regulated during the transition to a mature neuron (*Ballas et al., 2005)* and overexpression of REST by in utero electroporation leads to delay in neuronal maturation (*Mandel et al., 2011*). These results suggest that REST serves as a timer of terminal neuronal differentiation.

Counter to this model, in vivo loss-of-function analyses in mice have not shown evidence for precocious differentiation or de-repression of genes responsible for the mature neuronal phenotype in the embryonic nervous system. A global germline knockout (KO) of REST is early embryonic lethal, but no obvious morphological defects in neural tube formation were noted in that study

**eLife digest** In the brain, cells called neurons connect to each other to form complex networks through which information is rapidly processed. These cells start to form in the developing brains of animal embryos when "neural" stem cells divide in a process called neurogenesis. For this process to proceed normally, particular genes in the stem cells have to be switched on or off at different times. This ensures that the protein products of the genes are only made when they are needed.

Proteins called transcription factors can bind to DNA to activate or inactivate particular genes; for example, a transcription factor called REST inactivates thousands of genes that are needed by neurons. During neurogenesis, the production of REST normally declines, and some studies have shown that if the production of this protein is artificially increased, the formation of neurons is delayed. However, other studies suggest that REST may not play a major role in neurogenesis.

Here, Nechiporuk et al. re-examine the role of REST in mice. The experiments used genetically modified mice in which the gene that encodes REST was prematurely switched off in neural stem cells. Compared with normal mice, these mutant mice had much smaller brains that contained fewer neurons because the stem cells stopped dividing earlier than normal. Unexpectedly, many genes that are normally switched off by REST, were not significantly changed, while genes that are not normally regulated by REST – such as the gene that encodes a protein called p53 – were active.

It is known from previous work that p53 is expressed when cells are exposed to harmful conditions that can damage DNA. This helps to prevent cells from becoming cancerous. Nechiporuk et al. found that cells that lacked REST had higher levels of DNA damage than normal cells due to errors during the process of copying DNA before a cell divides. Furthermore, when both REST and p53 were absent, the neural stem cells became cancerous and formed tumors in the mice.

Nechiporuk et al.'s findings suggest that REST protects the DNA of genes that are needed for neurons to form and work properly. The new challenge is to understand where in the genome the damage is occurring.

(*Chen et al., 1998*). Later, in embryogenesis, brain-specific loss of REST, by conventional Cre lox technology, also lacks an obvious nervous system phenotype (*Aoki et al., 2012*), while conditional loss of REST from adult neural progenitors shows only transient and subtle precocious neuronal differentiation (*Gao et al., 2011*). Despite these results, a recent study points to a role for REST in human neurogenesis and microcephaly through regulation of REST by a factor, ZNF335, mutated in patients with a severe form of microcephaly (*Yang et al., 2012*). Additionally, microcephaly results from dysregulation of a REST/BAF170/Pax6 repressor complex during neurogenesis (*Tuoc et al., 2013*). Thus, the role of REST in embryonic neurogenesis remains an open question.

To re-examine the role of REST during embryonic neurogenesis, we use mice containing a conditional gene trap (GT) cassette in an intron of the endogenous *Rest* gene that terminates transcription upstream from the initiator codon. Using this line, we generate mice with a REST deficiency in nestin-positive neural progenitors, prior to the time when REST is dismissed normally from chromatin during neurogenesis. We examined these mice for embryonic and adult brain phenotypes and found DNA damage, apoptosis, and smaller brain size as prominent defects. The DNA damage persisted and caused glioblastoma (GBM) in mice also lacking the tumor suppressor, p53. We also characterized REST binding properties and embryonic phenotypes in a conventional brain-specific *Rest* KO line (*Gao et al., 2011*), targeting *Rest* exon 2, which we show still expresses a C-terminal REST peptide, for comparison with our *Rest GT* mice. Our results indicate that REST is required to protect genomic integrity, supervised by S phase surveillance, and that this function is key for regulating proper timing of terminal neuronal differentiation.

## Results

### Global Rest loss using a GT approach

We exploited a mouse line carrying a GT in the *Rest* intron (*Rest^{GT}*) between non-coding exon 1a–c and the first coding exon, exon 2 (*Figure 1A*). The *GT* cassette contains a splice acceptor site

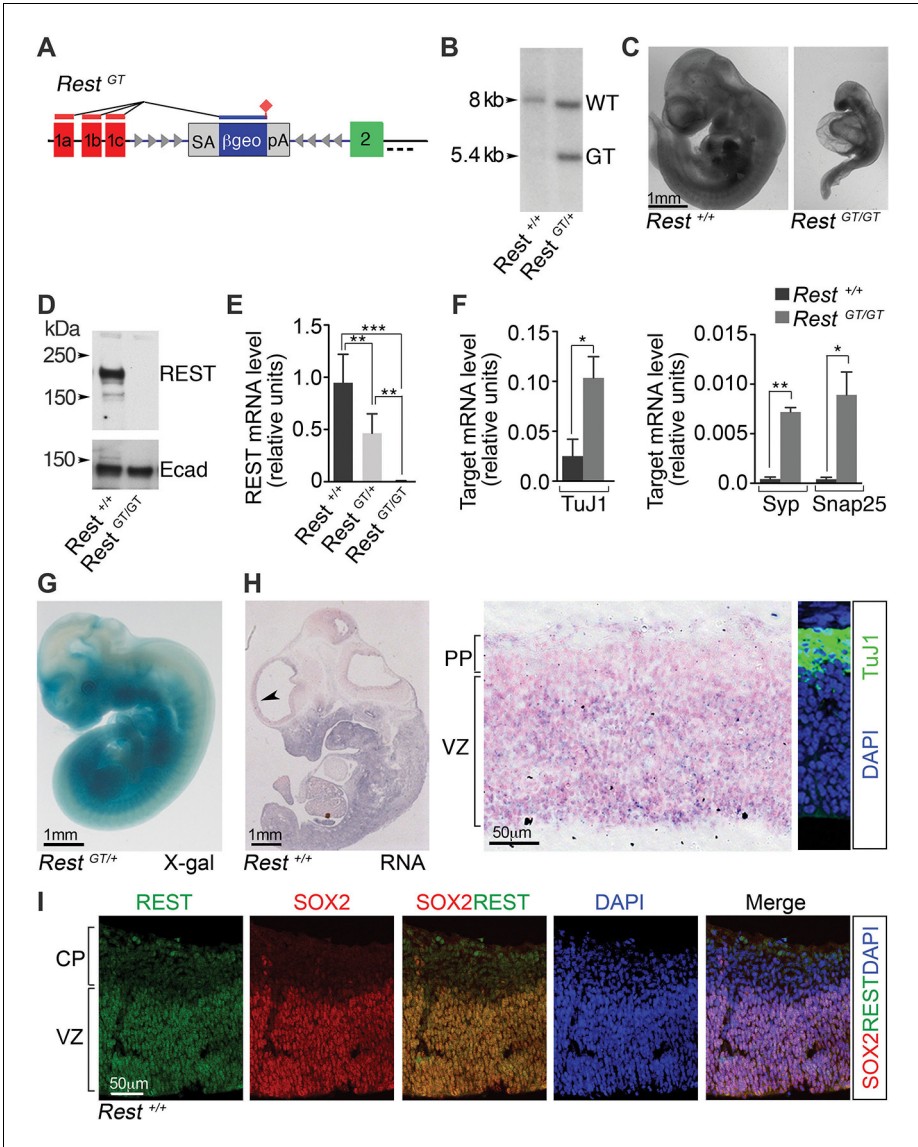

**Figure 1.** A GT in the *Rest* gene (*Rest^GT^*) causes REST deficiency and embryonic lethality. (**A**) Schematic showing the GT in the first *Rest* intron. Red and green boxes indicate alternative 5′ untranslated exons (1a–c) and first coding exon (2), respectively. The GT cassette contains an SA site, a reporter gene encoding a *β*-galactosidase neomycin fusion gene (*β-geo*), and a pA sequence. Arrowheads depict target sites for Flpe and Cre recombinases. Dashed line indicates probe location for Southern blot in B. (**B**) Southern blot of genomic DNA from indicated genotypes. (**C**) E10.5 wild type (*Rest^{+/+}^*) and mutant (*Rest^{GT/GT}^*) embryos. (**D**) Western blot analysis of REST protein in E10.5 embryos. Ecad (E-cadherin), loading control. (**E**) qRT-PCR analyses of *Rest* mRNA levels, normalized to 18S RNA, in E9.5 embryos, n=6 mice/genotype. Means and SD are shown. (**F**) qRT-PCR analysis, normalized to 18S RNA, for *Rest* targets, n=3 mice/genotype. Means and SD are shown. *Syp* and *Snap25* values in *Rest^{+/+}^* are $4.3 \times 10^{-4} \pm 2.0 \times 10^{-4}$ and $4.1 \times 10^{-4} \pm 2.0 \times 10^{-4}$, correspondingly. Statistical significance was determined by ANOVA with Tukey posthoc (**E**) and by unpaired t-test with Welch correction. (**F**) (**G**) Whole mount X-gal staining of E11.5 embryo. (**H**) Left, in situ hybridization analysis for *Rest* transcripts in E12.5 embryo. Arrowhead indicates region magnified in adjacent image. Counterstain (pink) is nuclear fast red. Right panel, Immuno-labeling showing location of TuJ1+ neurons used to determine PP and VZ boundaries. (**I**) Immuno-labeling of cortical section from E13.5 embryo using indicated antibodies and DAPI stain for nuclei. *, p<0.05, **, p<0.01, ***, p<0.001. ANOVA, analysis of variance; GT, gene trap; mRNA, messenger RNA; PP, preplate; qRT-PCR, quantitative real-time polymerase chain reaction; SA splice acceptor; SD, standard deviations; VZ, ventricular zone.

upstream of a promoter-less β-*galactosidase* and *neomycin* gene fusion (*β-geo*) and a polyadenylation sequence (*Schnutgen et al., 2005*). Thus, *β-geo* expression in this line is under the control of endogenous *Rest* regulatory elements. The GT was confirmed in the *Rest* locus by Southern blot analysis (*Figure 1B*) and we verified the insertion of a single GT in the genome by additional Southern blot and DNA sequence analysis (data not shown). *Rest*$^{GT/GT}$ mice, like *Rest* KO mice generated by germ-line deletions of *Rest* exons 2 and 4 (*Chen et al., 1998*; *Aoki et al., 2012*), are growth-arrested (*Figure 1C*) and die between E9.5 and 11.5, validating the *Rest*$^{GT}$allele as a model for loss-of-function.

In *Rest*$^{GT/GT}$ mice, we also observed loss of both REST protein (*Figure 1D*) and messenger RNA (mRNA; *Figure 1E*) (*Rest*$^{+/+}$, 0.95, standard deviation [SD], 0.27; *Rest*$^{GT/+}$, 0.47, SD 0.18; *Rest*$^{GT/GT}$, $0.7 \times 10^{-2}$, SD 0.003). This was accompanied by the predicted reciprocal up-regulation of select REST target genes (*Figure 1F*). The *β*-gal activity programmed from the GT correlated tightly with the pattern of endogenous *Rest* mRNA. For example, *β*-gal activity (*Figure 1G*) and endogenous *Rest* expression (*Figure 1H*, left panel) were both detected in non-neural tissues outside the developing nervous system. In the embryonic brain, endogenous *Rest* mRNA was confined largely to neural progenitors in the ventricular zone (VZ) and mostly absent from preplate cells that were populated with TuJ1+ neurons (*Figure 1H*). Correspondingly, REST protein expression was confined predominantly to SOX2+ neural progenitors (*Figure1I*). We were not able to detect REST protein in the subventricular zone (SVZ) occupied by the more committed TBR2+ basal progenitors (not shown), indicating that the down-regulation of REST occurred most robustly prior to the generation of mature neurons and the transition to basal progenitors.

## Conditional REST gene deficiency in neural progenitors

Early embryonic lethality, coincident with the onset of neurogenesis, precluded analysis of REST function in *Rest*$^{GT/GT}$ neural progenitors. Therefore, we used a two-step breeding scheme to remove REST specifically from neural progenitors. In the first step, *Rest*$^{GT}$ mice (*Figure 1A*) were crossed to mice expressing the *Flpe recombinase* transgene (*Dymecki et al., 2000*). This resulted in inversion of the GT cassette (*GTinv*) to restore normal splicing of *Rest* exons 1a–c to exon 2 (*Figure 2A*, top). In the second step, *Rest*$^{GTi/+}$ mice, heterozygous for the inverted allele, were bred to mice expressing a *Nestin Cre* recombinase transgene. This resulted in re-inversion of the *GTinv* cassette to create a mutant *Rest* allele in which exons 1a–c were spliced to the *β-geo* gene instead of exon 2 (*Figure 2A*, bottom), terminating transcription upstream of remaining *Rest* sequences. All genotypes were confirmed by DNA sequence analysis.

We examined β-gal activity in the developing brains of *Cre+, Rest*$^{GTi\ /GTi}$ and *Rest*$^{GTi/GTi}$ (controls hereafter) mice. Consistent with the global GT, β-gal activity matched expression of the endogenous *Rest* gene. Specifically, β-gal+ cells in the *Cre+, Rest*$^{GTi/GTi}$ mice were confined to neurogenic areas of E11.5 embryos (*Figure 2B*). Within the cortical VZ at E13.5, β-gal activity (*Figure 2C*) and endogenous *Rest* mRNA (data not shown) were detected primarily in the apical region, which is occupied by progenitors (*Götz and Huttner, 2005*). Interestingly, we also detected β-gal+ cells in some presumably mature cells in the marginal zone (MZ) of the cortical plate (CP) indicating *Rest* promoter activity in a subset of neurons (*Figure 2C*).

Ninety-five percent of the *Cre+, Rest*$^{GTi\ /GTi}$ mice survived into adulthood, but had significantly reduced REST protein and mRNA levels in neural progenitors and E13.5 brains compared with controls (*Figure 2D and E*). Chromatin immunoprecipitation (ChIP) analysis of E13.5 brains from *Cre+, Rest*$^{GTi\ /GTi}$ mice indicated ~four-fold reduction in REST occupancy at consensus RE1 sites within 1kb of the *Glycine receptor* and *Snap25* transcriptional start sites, in the first exon and intron, respectively (*Glycine receptor: Rest*$^{GTi\ /GTi}$, 0.64, SD 0.13; *Cre+, Rest*$^{GTi\ /GTi}$, 0.16, SD 0.07), *Snap 25: Rest*$^{GTi\ /GTi}$, 0.32, SD 0.04; *Cre+, Rest*$^{GTi\ /GT}$, 0.07, SD 0.04) (*Figure 2F*). There was no significant change in REST occupancy in the *Snap25* coding sequence or the *myf5* promoter region that lacked RE1 binding sites (*Figure 2F*). The very low levels of glycine receptor mRNA in control mice was not within the linear range of detection. Levels of this mRNA in the mutant, normalized to 18S RNA, were consistently within the linear range, but still very low (0.13, SD 0.08). For SNAP25, we observed an ~2-fold increase in mutant mRNA compared with control levels (*Rest*$^{GTi/GTi}$: 0.09, SD 0.06; *Cre+, Rest*$^{GTi/GTi}$: 0.2, SD 0.05; p <0.05, unpaired t-test, n=7–8, E13.5 brain). It is possible that the low mRNA levels for both genes reflect the absence of transcriptional activators at this time.

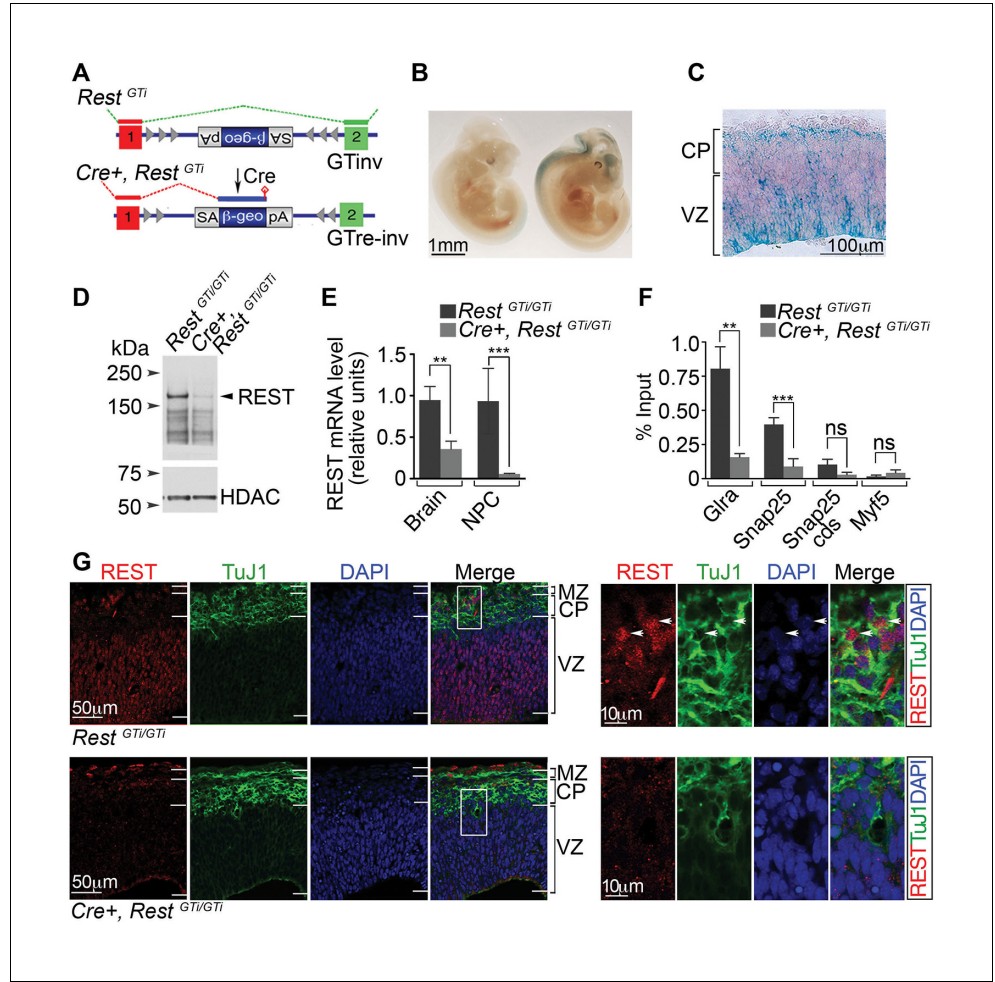

**Figure 2.** A conditional *Rest^GT* allele results in REST-deficiency in Nestin+ progenitors. (**A**) KO strategy. Top, mice bearing an inverted GT cassette (*Rest^GTinv*), resulting in normal splicing, were generated by mating *Rest^GT* mice (*Figure 1A*) to mice containing the *Flpe* transgene. Bottom, conditional mutants, resulting in splicing of exon 1 to the GT cassette (*Cre+, Rest^GTi*), were generated by mating *Rest^GTi* mice (top) to mice bearing the *Nestin Cre* transgene. It is noteworthy that *β-geo* expression was still under the control of *Rest* regulatory elements. (**B**) Whole-mount X-gal staining of E11.5 *Rest^GTi/+* (left) and *Cre+, Rest^GTi/+* (right) embryos. (**C**) X-gal staining of a coronal section of E13.5 cortex from *Cre+, Rest^GTi/+* mice. (**D**) Western blot of nuclear extracts from neuroepithelia of E13.5 mice. HDAC, histone deacetylase 2, loading control. (**E**) qRT-PCR analysis of *Rest* transcripts, normalized to 18S RNA, in E13.5 brain (n=6 mice/genotype) and NPCs grown as neurospheres for 5 days (n=3 mice/genotype). (**F**) Quantitative chromatin immunoprecipitation analysis of REST enrichment at RE1 sites in the glycine receptor (*Glra*; +275 bp from TSS) and *Snap25* genes (+867 bp from TSS) (n=4 mice/genotype). Amplicons were designed within 100 bp of RE1 binding sites. *Snap25* CDS and *Myf5* transcriptional start site lack RE1 sites. Means and SD are shown in (**E**) and (**F**). Statistical significance was determined by Mann–Whitney test in **E** and by unpaired t-test with Welch correction in **F**. (**G**) Immuno-labeling of E13.5 telencephalon using indicated antibodies. Boxes indicate regions of higher magnification in images at right. White arrowheads indicate DAPI+ cells that express both proteins. *, p<0.05, **, p<0.01, ***, p<0.001. CDS, coding sequence; CP, cortical plate; GT, gene trap; KO, knockout; MZ, marginal zone; NPC, neural progenitor cells; qRT-PCR, quantitative real-time polymerase chain reaction; SD, standard deviation; VZ, Ventricular zone.

In control mice, SOX2 progenitors of the VZ stained positively for REST, but the majority of TuJ1 + neurons were not immuno-positive (*Figure 1I and 2G*), consistent with the known down-regulation of REST during neurogenesis (*Ballas et al., 2005*). Interestingly, however, we could detect REST protein in a small number of TuJ1+ neurons in the outmost MZ of the CP (*Figure 2G*, right images), corroborating β-gal staining (*Figure 2C*), suggesting either that REST is re-expressed at later stages of

differentiation or that REST expression has not yet been extinguished completely in these neurons. Whether REST is bound to the RE1 sequence in the TuJ1 gene at this stage cannot be determined given the small number of labeled neurons.

## Cre+, Rest$^{GTi/GTi}$ mutants have smaller brains, thinner cerebral cortices, and reduced numbers of upper layer neurons

Because a previous study indicated that loss of REST was associated with microcephaly (*Yang et al., 2012*), we asked whether this was also true for our *Rest$^{GT}$* mice. Indeed, significantly smaller brains were evident at birth (data not shown) and in postnatal (P45) *Cre+, Rest$^{GTi/GTi}$* mice when compared with control mice of two different control genotypes, *Rest$^{GTi/GTi}$* and Nestin Cre (*Figure 3A and B* left panel). The brain size in the mutant was reduced to 71% of the brain size of *Rest$^{GTi/GTi}$* mice (*Rest$^{GTi/GTi}$*, 0.46, 95% confidence interval [CI] 0.45–0.46; *Cre+, Rest$^{GTi/GTi}$* 0.33, CI 0.32–0.34), similar to the difference from *Nestin Cre* mice (*Nestin Cre+, Rest$^{+/+}$*: 0.43, CI 0.42 to 0.44). A third potential control line is *Cre+, Rest$^{GTi/+}$* mice. However, we measured a significant brain size reduction *Cre+, Rest$^{GTi/+}$* mice (0.36, CI 0.35–0.37, p value <0.001) compared to *Rest$^{GTi/GTi}$* (reduction to 78%) or *Cre+, Rest$^{+/+}$*(reduction to 83%) mice, likely due to the combination of the slightly reduced brain size of the Cre recombinase background and reduced REST levels (*Figure 3B* and *1E*). The hypomorphic effect is consistent with previous studies, indicating that gene expression levels are sensitive to small changes in REST levels (*Ballas et al., 2005*; *Ballas and Mandel, 2005*), so the *Rest$^{GTi/GTi}$* mice were used as controls in the remaining experiments.

We also tested inversion of the *Rest$^{GTi}$* allele using mice expressing an *hGFAP Cre* recombinase transgene active at mid-neurogenesis, slightly later than *Nestin Cre* (*Zhuo et al., 2001*). In these mice, there was a smaller reduction in brain mass to only 92% of control levels (*Figure 3B* right panel), suggesting a critical temporal window for REST function at early-to-mid neurogenic stages. This matches in utero electroporation experiments revealing an enhanced migration phenotype of REST knockdown performed at E13.5 that is not observed at E14.5 (*Yang et al., 2012*; *Fuentes et al., 2012*).

The reduction in brain mass in REST mutant mice correlated with reduced cortical thickness, both rostrally and caudally, and reduced corpus callosum thickness (*Figure 3C and D*). To determine whether these phenotypes were associated with an imbalance of temporally distinct progenitor types, we counted neurons in the six cortical layers that are born at different times during neocortical development, with deep layer neurons preceding the birth of upper layer neurons. To this end, we performed dual immunostaining for transcription factor markers specific for adjacent layers (*Alcamo et al., 2008*; *Britanova et al., 2008*; *Arlotta et al., 2005*; *Bedogni et al., 2010*). There was a reduction to 56% of control numbers of SATB2+/CTIP2- upper layer 2–4 neurons (*Rest$^{GTi/GTi}$*, 769.3, CI 733.2 – 805.4; *Cre+, Rest$^{GTi/GTi}$*: 428.4, CI 397.2–459.5), but no statistically significant differences between mutant and control in layer 5 and 6 neurons that were born earlier in neurogenesis (*Figure 3E,F and G*). However, CTIP2 immuno-labeling, which molecularly defines layer 5, showed expansion into layer 6 and decreased density in mutant brain (*Figure 3F and H*), pointing to some disorganization due to the premature loss of REST. Despite this finding, the predominant feature of loss of REST during neurogenesis was significantly fewer postnatal neurons in the upper cortical layers born during mid-to-late neurogenesis.

## Premature loss of REST from neural progenitors leads to premature cell cycle exit

The small brain size and diminished numbers of neurons at birth could reflect increased depletion of progenitor cells and/or cell death. To address the former possibility, we first distinguished apical and basal progenitors by immuno-labeling with antibodies to PAX6 and TBR2, respectively (*Figure 4A*). Numbers of PAX6+TBR2− apical progenitors in *Cre+ Rest$^{GTi/GTi}$* mice were reduced to ~71% of control values (*Rest$^{GTi/GTi}$*, 273.1, CI 203.4–342.9; *Cre+, Rest$^{GTi/GTi}$*: 123.7, CI 110.8.2–136.5; *Figure 4B*). There were also reduced numbers of TBR2+ basal progenitors, which are progeny of the apical PAX6+ progenitors, at E13.5 (*Rest$^{GTi/GTi}$*, 116.0, CI 100.2–131.8; *Cre+, Rest$^{GTi/GTi}$*: 81.9, CI 62.2–101.5) (*Figure 4B*).

To determine whether the depletion of apical progenitors correlated with premature cell cycle exit, we pulsed E13.5 embryos with BrdU to label cells that were undergoing DNA synthesis. After

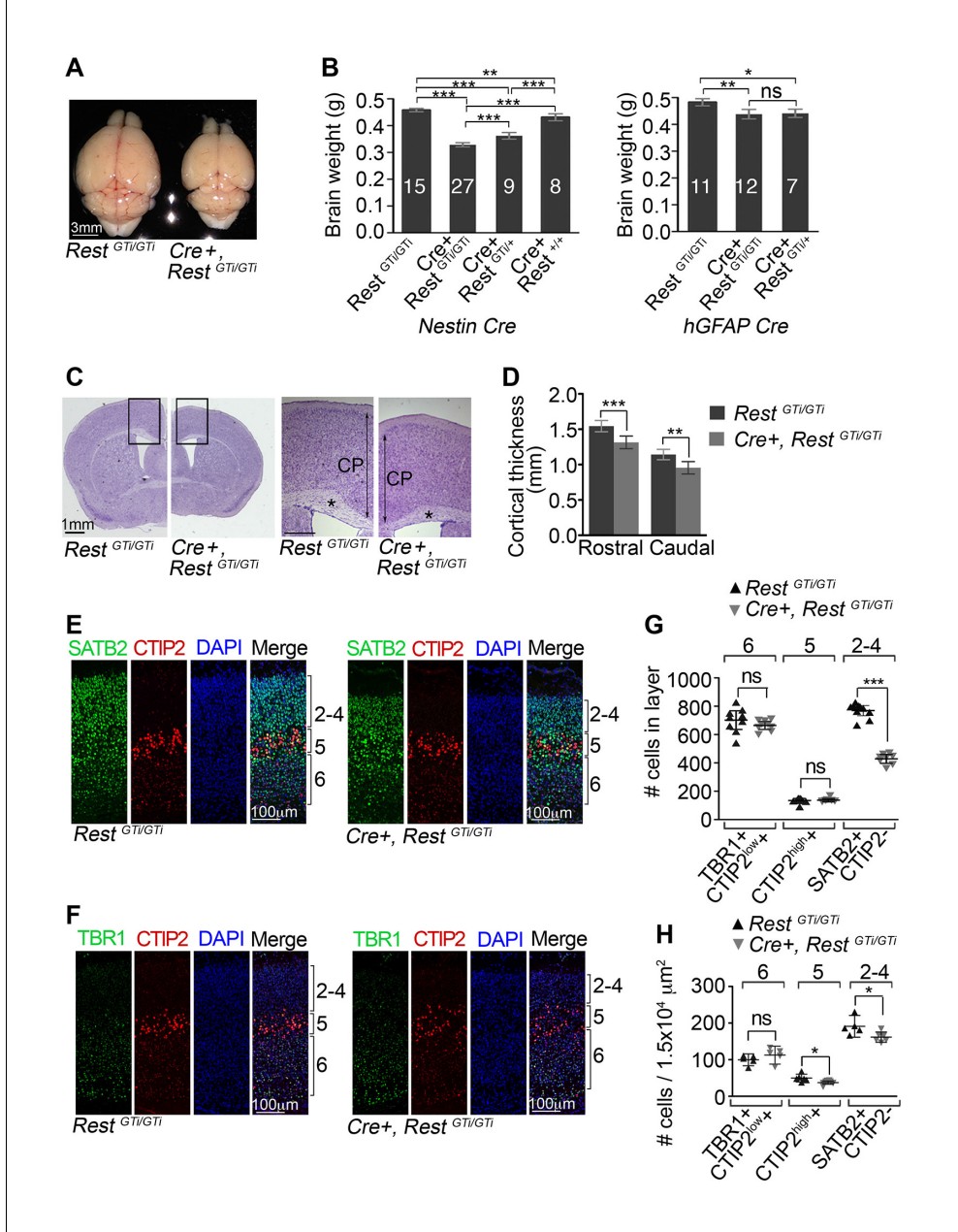

**Figure 3.** Reduced brain size, thinner cortex, and reduced numbers of upper layer neurons in *Nestin Cre+,* *Rest*$^{GTi/GTi}$ mice. (**A**) Representative P45 brains from control and *Cre+, Rest*$^{GTi/GTi}$ littermates. (**B**) Comparison of brain mass in P45 mice deleted for REST using *Nestin* (left panel) and *hGFAP* (right panel) promoter driven Cre recombinases. The numbers of mice analyzed for each genotype are shown. Statistical significance determined by ANOVA with Tukey posthoc (left) and Kruskal–Wallis ANOVA test with Dunn posthoc (right). (**C**) Nissl-stained P45 coronal brain sections. Boxes denote higher magnification views in right hand panels. *, corpus collosum. Scale bar for right panel, 0.5mm. (**D**) Measurements of cortical thickness in P45 brain (n=8–11 mice/genotype). Statistical significance determined by ANOVA with Tukey posthoc. (**E**) and (**F**) Representative immuno-labeling of P1 cortical sections with indicated antibodies revealing cortical layering in control and *Cre+, Rest*$^{GTi/GTi}$ mice. The presence of both low and high expressing CTIP2 cells in layers 6 and 5, respectively, is noteworthy. (**G**) Quantification of (**E**) and (**F**) numbers over brackets denote cortical layers where the cells were counted. Cells were counted in 400 μm of cortical thickness (n=8–10 mice/genotype). Note that only 200 μm images are shown in E, F. Low (layer 6) and high (layer 5) CTIP2-expressing cells were used to differentiate between Tbr1+, CTIP2$^{low}$+ cells of layer 5 and CTIP$^{high}$+ cells in layer 5. SATB2+ CTIP2- cells were counted above layer of CTIP$^{high}$+ cells. (**H**) Mean cell densities from three areas (100×150 μm$^2$) in each cortical section, n=5–8 mice/genotype. Statistical significance was

*Figure 3. continued on next page*

*Figure 3. Continued*
determined by Mann–Whitney t-test for (**G**) and (**H**). Means and 95% CI are shown in B, D, F and G. *, p<0.05; **, p<0.01; ***, p<0.001. ANOVA, analysis of variance; ns, non-significant.

24 hr, we immunolabeled for Ki67, a marker of cycling cells, and incorporated BrdU, and quantified the results relative to the number of DAPI+ nuclei. The Ki67 staining in *Cre+, Rest*$^{GTi/GTi}$ cells was reduced to 69% of the control values (*Rest*$^{GTi/GTi}$, 72%, CI 70–73%; *Cre+, Rest*$^{GTi/GTi}$, 50%, CI 38–40%), with no change in the percentage of BrdU+ cells (*Rest*$^{GTi/GTi}$, 42%, CI 37–48%; *Cre+, Rest*$^{GTi/GTi}$, 43%, CI 38–49%) (*Figure 4C and D*, top panel). To determine the fraction of cells that exited the

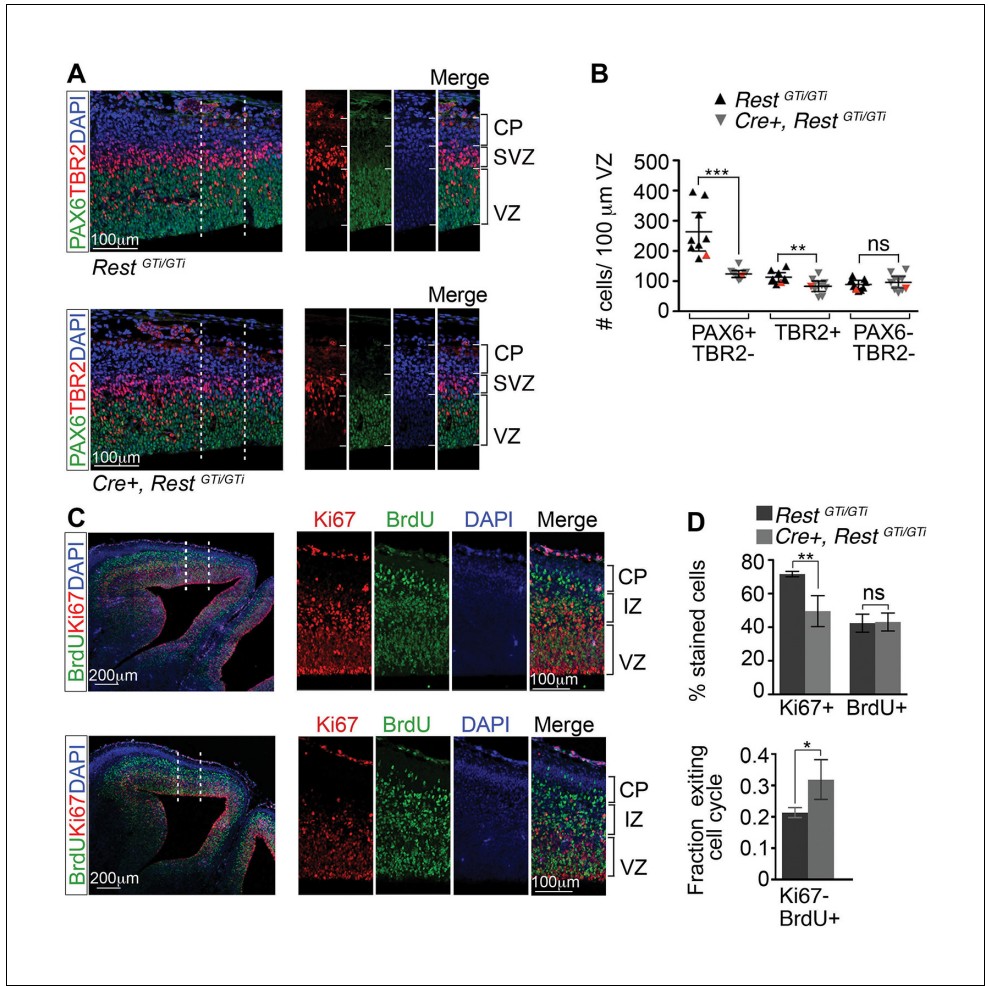

**Figure 4.** Depletion of apical progenitors and premature cell cycle exit in *Cre+, Rest*$^{GTi/GTi}$ mice. (**A**) Representative immuno-labeling distinguishing apical (PAX6+/TBR2−) and basal (TBR2+) progenitors in E13.5 cortices. Area outlined by dotted lines is shown in images at right. (**B**) Quantification of A. Measurements from 100 µm cortical width extending from the VZ to pial surface, n=9 mice /genotype. Statistical significance was determined by Mann–Whitney t-test. Red triangles show quantification of the images shown in A. (**C**) Representative immuno-labeling for Ki67 and BrdU in the cortical section of E14.5 mouse brain pulsed with BrdU for 24 h. The area outlined by dotted lines is magnified in the adjacent images. (**D**) Top, histogram showing percent stained cells relative to the number of DAPI+ nuclei. Bottom, fraction of cells exiting the cell cycle defined as fraction of Ki67-BrdU+ cells in total BrdU+ cell population. Measurements were made in 200 µm cortical width areas extending from ventricular to pial surface, n=7 mice/genotype. Statistical significance was determined by Mann–Whitney t-test. Means and 95% CI are shown in **B** and **D**. *, p<0.05; **, p<0.01; ***p<0.001. IZ, intermediate zone; SVZ, subventricular zone.

cell cycle in a 24 hr period, we quantified the proportion of Ki67- cells in the total BrdU+ cell population. Our results indicated that between E13.5 and 14.5, the progenitor pools were increasingly depleted, as ~50% more progenitors exited the cell cycle in *Cre+, Rest$^{GTi/GTi}$* compared with controls (*Rest$^{GTi/GTi}$*, 21%, CI 20–23%; *Cre+, Rest$^{GTi/GTi}$*, 32%, CI 26–38%) (*Figure 4D*, lower panel). This indicates a decreasing progenitor pool available to generate the late born upper layer neurons (*Figure 3E and G*).

## REST-deficient neural progenitors undergo p53-mediated cell death linked temporally to distinct neuronal differentiation programs

Microarray analysis of E12.5 brain from *Cre+, Rest$^{GTi/GTi}$* and control mice did not show significant up-regulation (>1.3 fold) in canonical REST neuronal target genes (*Supplementary file 1*). However, our analysis did reveal significant up-regulation of microglial signature genes (*Hickman et al., 2013*; and several p53-regulated pro-apoptotic genes (*Ko and Prives, 1996*; *Levine, 1997*; *Budanov and Karin, 2008*) in brain tissue and LeX+ sorted progenitors (*Supplementary file 1* and *Figure 5A*). To test for cell death, we stained E13.5 brain sections from the cortex and lateral ganglionic eminence (LGE) with antibody against the activated form of cleaved Caspase3 (ClCasp3), a member of the cysteine–aspartic acid proteases family, which is a critical mediator of apoptosis and required for chromatin condensation and DNA fragmentation (*Janicke, 1998*). Unlike in control cortices, apoptosis was prominent in cells in *Cre+, Rest$^{GTi/GTi}$* mice, particularly at the border between the VZ/SVZ and CP (*Figure 5B*). Interestingly, although cells in the VZ/SVZ border area are densely populated with TBR2+ basal progenitors, the apoptotic cells were not positive for TBR2+ (*Figure 5—figure supplement 1A*).

Cell death peaked at E14.5 in the cortex, the time of mid-neurogenesis for this brain region and the beginning of upper layer neuronal specification (*Caviness et al., 2009*), and then declined to control levels by E15.5 (*Figure 5C*). The peak of apoptosis was earlier in LGE, peaking at E13.5 (*Figure 5C*). The number of clCasp3+ cells in *Rest$^{GTi/GTi}$* LGE was similar to the low numbers in the cortex (not shown). The distinct peak times of apoptosis for the cortex and LGE, coincident with their staggered time courses of differentiation (*Batista-Brito and Fishell, 2009*; *Greig et al., 2013*; *Colasante and Sessa, 2010*) indicate that apoptosis was linked to the timing of premature cell cycle exit of distinct progenitor populations in these brain regions, and not to a nonspecific effect on all progenitors or effects on earlier-stage, less-committed neural progenitors. The majority of clCasp3+ cells were also positive for MAP2 and negative for Ki67 (*Figure 5D* and *Figure 5—figure supplement 1B*). Because MAP2 is not a direct REST target gene, this result suggests progression of a mature neuronal program, but the tight correlation between MAP2 expression and apoptosis prevents knowledge of the sequence of events.

To confirm that the apoptosis was due to activation of pro-apoptotic p53 pathways indicated by the microarray analysis, we generated compound conditional mutant mice that lacked both the *Rest* and *Trp53* genes (*Jonkers et al., 2001*) in *Nestin Cre+* cells and their progeny (*Cre+, Rest$^{GTi/GTi}$, Trp53$^{fl/fl}$*). At E13.5, numbers of apoptotic cells/100μm VZ width in the compound mutant were rescued to control levels (*Rest$^{GTi/GTi}$*), in both the cortex (*Figure 5E*) (*Rest$^{GTi/GTi}$*, 0.7, CI 0–0.4; *Cre+, Rest$^{GTi/GTi}$, Trp53$^{fl/fl}$*: 2.9, CI 1.5–4.4; *Cre+, Rest$^{GTi/GTi}$*, 49.1, CI 39.5 –-58.6) and the LGE (*Rest$^{GTi/GTi}$*: 0.07, CI 0.0–0.3; *Cre+, Rest$^{GTi/GTi}$, Trp53$^{fl/fl}$*, 0.53, CI 0.29 –-0.78; *Cre+, Rest$^{GTi/GTi}$*, 128.7, CI 107.8–149.5). The mRNA levels of elevated p53 pro-apoptotic transcriptional targets were also reduced to control levels (*Figure 5F*). Brain size was restored to only 78% of control values (*Figure 5G*), suggesting that progenitor pool depletion by both premature cell cycle exit and apoptosis contribute to this phenotype. It was possible that microarray analysis showing lack of induction of neuronal genes in *Cre+, Rest$^{GTi/GTi}$* mice (*Supplementary file 1*) was due to the elimination of many cells by apoptosis. Even after eliminating apoptosis, however, microarray analysis still did not show significant up-regulation (>1.5-fold over control) of REST-regulated mRNAs (*Supplementary file 2* and *Figure 5—figure supplement 2*), suggesting that premature differentiation per se was not the sole underlying cause of the apoptosis.

GT alleles carrying *β-geo* cassettes have been used extensively for both global and conditional gene inactivation in mice, with no reports of *β-geo* toxicity or chromosomal damage during recombination of the GT allele (*Budanov and Karin, 2008*; *Gossler et al., 1989*; *Skarnes et al., 1992*; *Zambrowicz et al., 2003*; *Krechowec et al., 2012*; *Mao et al., 1999*; *Petkau et al., 2012*; *Theroux et al., 2007*; *Peralta et al., 2014*; *Ishizawa et al., 2011*). Similarly, gene recombination

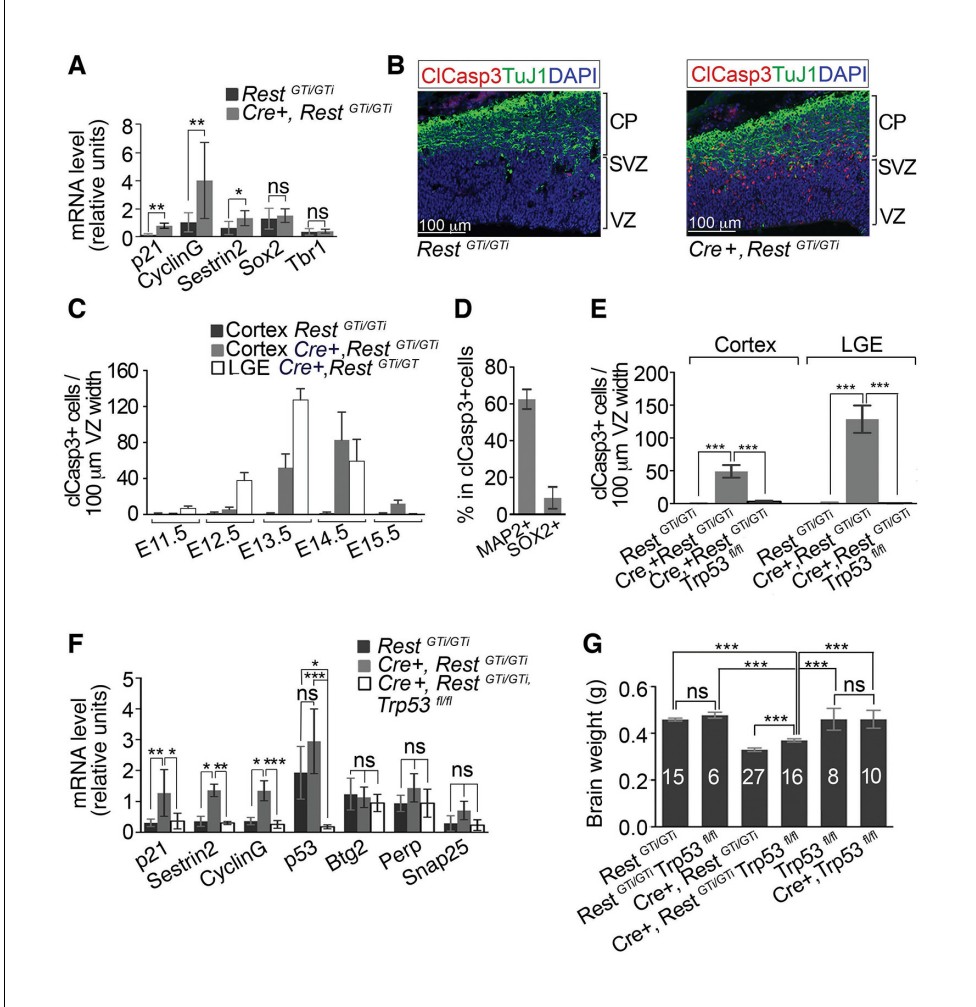

**Figure 5.** Progenitors and neurons in cortex of *Cre+, Rest*<sup>GTi/GTi</sup> mice undergo apoptosis that is rescued by deletion of *p53*. (**A**) qRT-PCR analyses normalized to 18S RNA, of p53 pro-apoptotic target mRNAs (*p21, Sestrin2 and CyclinG*), progenitor mRNAs (*Sox2*) and neuronal mRNAs (*Tbr1*) in LeX+ purified progenitors isolated from E13.5 brain. *Rest*<sup>GTi/GTi</sup>, n=4 mice, *Cre+, Rest*<sup>GTi/GTi</sup>, n=6 mice. (**B**) Representative immuno-labeling for apoptotic cells with clCasp3 and neuronal marker TuJ1 in coronal telencephalic sections from E13.5 mice. (**C**) Temporal profiles of apoptosis in cortex and LGE of *Cre+, Rest*<sup>GTi/GTi</sup> mice and cortex of *Rest*<sup>GTi/GTi</sup> mice, in areas of 100 μm ventricular width extending from VZ to the pial surface (n=6–9 mice/time point). Means and SDs are shown. (**D**) Percentage of MAP2+ and SOX2+ cells in all apoptotic cells in E13.5 Cortex of *Cre+, Rest*<sup>GTi/GTi</sup> mice in 100μm ventricular width. n=5 mice. (**E**) Quantification of apoptosis in E13.5 cortex and LGE (n=5–9 mice/genotype/100μmVZ). Note: The same data for *Cre+, Rest*<sup>GTi/GTi</sup> cortex and LGE from 5C is re-plotted for comparison. (**F**) qRT-PCR analysis of mRNA levels, relative to 18S RNA, of p53 pro-apoptotic (*p21, CyclinG, Sestrin2, Perp*), non-apoptotic (*Btg2*) and non p53 (*Snap25*) targets in E12.5 brain (n=7–8 mice/genotype). (**G**) Measurements of brain mass in P45 mice. Numbers of mice are indicated in the histogram. Note: the same data from *Cre+, Rest*<sup>GTi/GTi</sup> and *Rest*<sup>GTi/GTi</sup> in *Figure 3B* is re-plotted for comparison. Means and 95% CI are shown in A, D, E, F, G. Statistical significance was determined by Mann–Whitney t-test (**A**), ANOVA test with Tukey posthoc (**E** and **G**) and Kruskal–Wallis ANOVA test with Dunn posthoc (**F**). *, p<0.05, **, p<0.01, ***, p<0.001, ns, non-significant. ANOVA, analysis of variance; clCasp3, cleaved caspase3; CI, confidence interval; LGE, lateral ganglionic eminence; mRNA, messenger RNA; qRT-PCR, quantitative real-time polymerase chain reaction; SD, standard deviation

The following figure supplements are available for figure 5:

**Figure supplement 1.** Apoptotic cells are TBR2− and primarily post-mitotic, MAP2+ cells.

**Figure supplement 2.** Volcano plot of microarray analysis comparing transcriptome of E12.5 brains of control, *Rest*<sup>GTi/GTi</sup>, *Trp53*<sup>fl/fl</sup>, and *Cre+, Rest*<sup>GTi/GTi</sup>, *Trp53*<sup>fl/fl</sup> mice (*Supplementary file 2*).

**Figure supplement 3.** qPCR analyses of DNA in the *Rest*<sup>GT</sup> genomic locus normalized to *Gapdh* DNA.

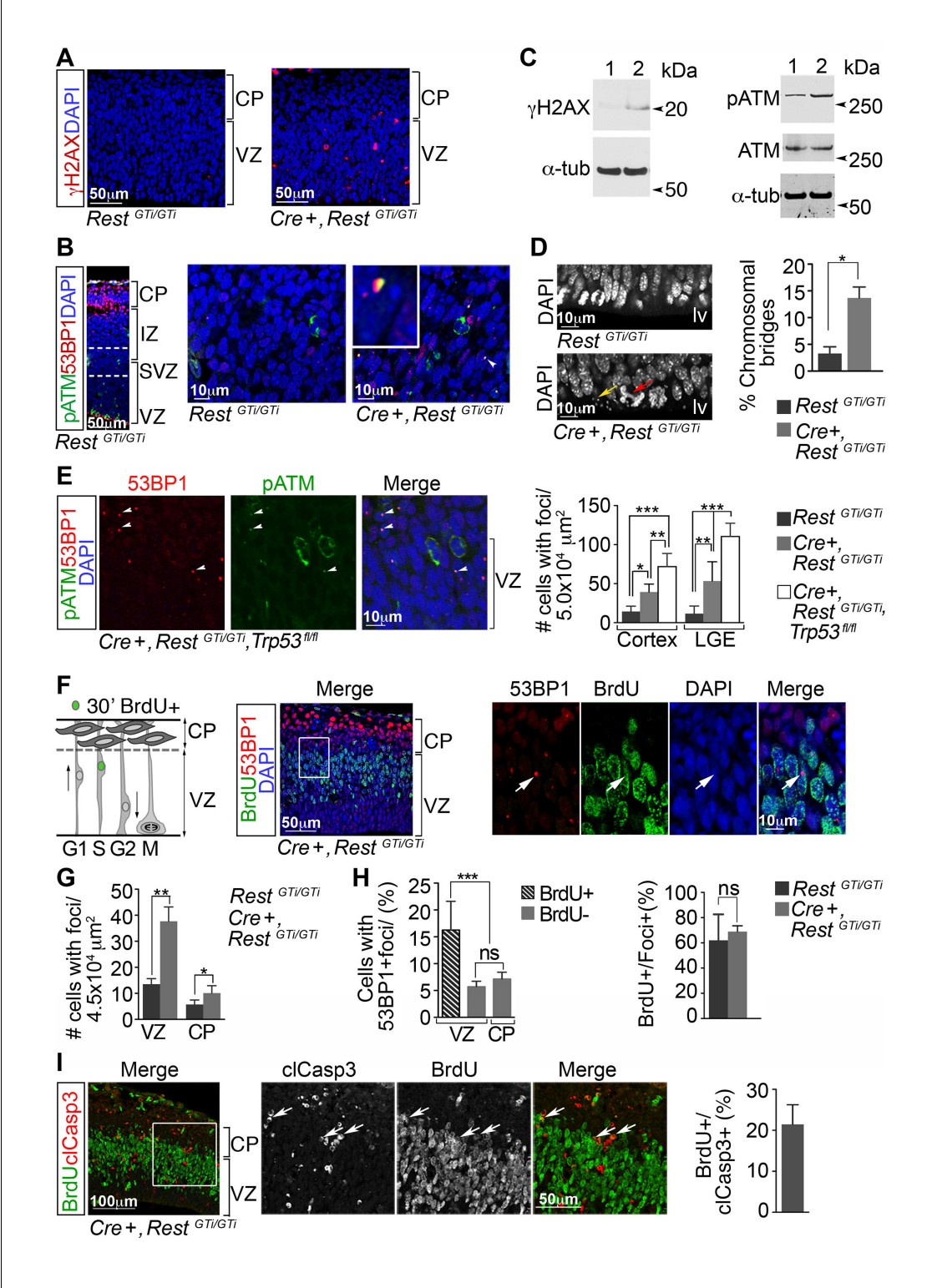

**Figure 6.** DNA damage in neural progenitors is due to loss of REST and persists in progenitors with loss of REST and p53. (**A**) Representative immuno-labeling of E13.5 cortex for phosphorylated histone protein H2AX (γH2AX). (**B**) Representative immuno-labeling for pATM and p53 binding protein 1 (53BP1) from cortex of E14.5 *Rest*$^{GTi/GTi}$ mice. The region between the dotted lines is shown at higher magnification in the right panels. Arrowhead indicates the nuclear foci magnified in inset. (**C**) Western blot analysis of E13.5 brain protein lysates. Lane 1, *Rest*$^{GTi/GTi}$; Lane 2, *Cre+, Rest*$^{GTi/GTi}$. α-tub (tubulin), loading control. (**D**) Left, Representative image of nuclei in cells in E13.5 ventricular surface. Red arrow points to abnormal bridge-like structures in mutant mice. Yellow arrow points to micronuclei. Right, quantification from E13.5 cortex. n=4 mice/genotype/100 mitoses. (**E**) Left,
*Figure 6. continued on next page*

Figure 6. Continued

immuno-labeling of VZ cells in E13.5 cortex. Right, quantification from cortex and LGE dual labeled for 53BP1 and pATM, n=5 mice/genotype, 3–5 sections/mouse. (**F**) Left, schematic showing nuclear positions of neural progenitors in S phase labeled by 30 m pulse of BrdU. Right, representative images from E13.5 cortical section after 30 m BrdU pulse. Cells in inset are shown at higher magnification. Arrowheads indicate DNA damage foci in cells that incorporated BrdU in the S phase. (**G**) Quantification of 53BP1 data represented in F. n=6 mice, 3–4 sections/mouse. (**H**) Quantification of cells with 53BP1 foci in E13.5 *Cre+, Rest^GTi/GTi* brain sections (**I**) Left, representative images of immunostained E13.5 cortical section. Cells in inset are shown at higher magnification. Right, quantification, n=6 mice, 3–4 sections/mouse/200μmVZ. Means and SD are shown. Statistical significance in **D** and **G** was determined by Mann-Whitney t-test and by ANOVA test with Tukey posthoc in **E** and **H**. *p<0.05; **p<0.01; ***p <0.001. ANOVA, analysis of variance; pATM, phosphorylated ataxia telangiectasia mutated; SD, standard deviation; VZ, ventricular zone.

defects using Cre recombination technology to stochastically invert tandem copies of a reporter gene have also not been reported (*Livet et al., 2007*). Nevertheless, to rule out this possibility for the *Rest^GT* allele, we compared relative amounts of *Rest* genomic and GT cassette DNA isolated from E13.5 brains of *Rest^GTi/GTi*, *Trp53^fl/fl* and *Cre+, Rest^GTi/GTi*, and *Trp53^fl/fl* mice. Using primers specific to these two regions for qPCR, we found no evidence for loss of DNA with inversion of the GT allele by Cre recombinase activity (*Figure 5—figure supplement 3*).

## Loss of REST function alone is responsible for DNA damage

Activation of the p53-pro-apoptotic pathway is often initiated due to genotoxic stress (*Ko and Prives, 1996*; *Levine, 1997*; *Budanov and Karin, 2008*; *Elledge and Zhou, 2000*). Therefore, we tested for DNA damage in the cortex of *Cre+, Rest^GTi/GTi* mice by staining for phosphorylated H2AX histone (γH2AX), an established marker for DNA damage (*Figure 6A*), and the presence of DNA damage nuclear foci, revealed by co-staining of phosphorylated ataxia telangiectasia mutated (pATM) kinase and p53 binding protein 1 (53BP1) (*Meek and Anderson, 2009*) (*Figure 6B*). While we observed γH2AX+ cells and cells with DNA damage nuclear foci in the cortices of E13.5 *Cre+, Rest^GTi/GTi* mice, we detected foci rarely in sections from controls. Western blot analysis verified the increased amounts of the pATM and γH2AX in the brains of E13.5 mutant mice (*Figure 6C*). In addition to molecular evidence for activation of a DNA damage-signaling cascade, we also observed an increased incidence of fragmented DAPI+ micronuclei and abnormal chromosomal bridges at the apical edge of the VZ in E13.5 *Cre+, Rest^GTi/GTi* mice when compared with controls, indicating abnormal separation of sister chromatids and DNA breakage (*Figure 6D*). Cells with DNA damage were eliminated by apoptosis, because loss of REST alone, in E13.5 cortex and LGE, resulted in fewer cells with DNA damage foci than in *Cre+, Rest^GTi/GTi*, *Trp53^fl/fl* mice (*Figure 6E*).

DNA damage responses often occur in association with defects in DNA replication, culminating in cell death. To test for this association, we pulsed E13.5 *Cre+, Rest^GTi/GTi* and control embryos for 30 min with BrdU, to label cells in S phase, and co-stained BrdU+ progenitors and neurons with antibodies to 53BP1 and clCasp3 for damage response foci and apoptosis, respectively (*Figure 6F–H*). At this developmental stage, due to interkinetic nuclear migration, a majority of cells in S phase are found in the VZ close to the border of the CP (*Ueno et al., 2006*; *Baye and Link, 2007*). There were significantly more cells with 53BP1 foci in the VZ and CP in *Cre+, Rest^GTi/GTi* brain compared with control (*Figure 6G*). Moreover, there were more cells with 53BP1 foci in the BrdU+ population in the mutant than in the BrdU− populations (*Figure 6H*, left). As expected, the majority of cells with 53BP1+ foci were also positive for BrdU in brains of both genotypes (*Figure 6H*, right). Analysis

**Table 1.** Glioma incidence and grade according to *Rest* and *Trp53* genotypes

| | *Cre+, Rest^GTi/GTi, Trp53^+/+* | *Cre+, Rest^GTi/GTi, Trp53^fl/fl* | *Cre+, Rest^GTi/+ Trp53^fl/fl* | *Cre+, Rest^+/+ Trp53^fl/fl* |
|---|---|---|---|---|
| Incidence | 0(30)* | 66%(131) | 53%(34) | 19%(68) |
| Grade I–III | - | 46% | 94% | 77% |
| Grade IV (GBM) | - | 54% | 6% | 23% |

* Number in parenthesis indicates number of animals used in analyses.

of 53BP1 foci+ cells in the clCasp3+ population in mutant mice revealed a time lag between S-phase damage and apoptosis, because only ~20% of cells labeled with clCasp3+ cells were also positive for BrdU (*Figure 6I*). We did not perform similar counts in control cortex because of the sparse numbers of clCasp3+ cells (2.4 cells, CI 03–4.4/ 200μm VZ width, n=4 animals, 3–4 sections per mouse).

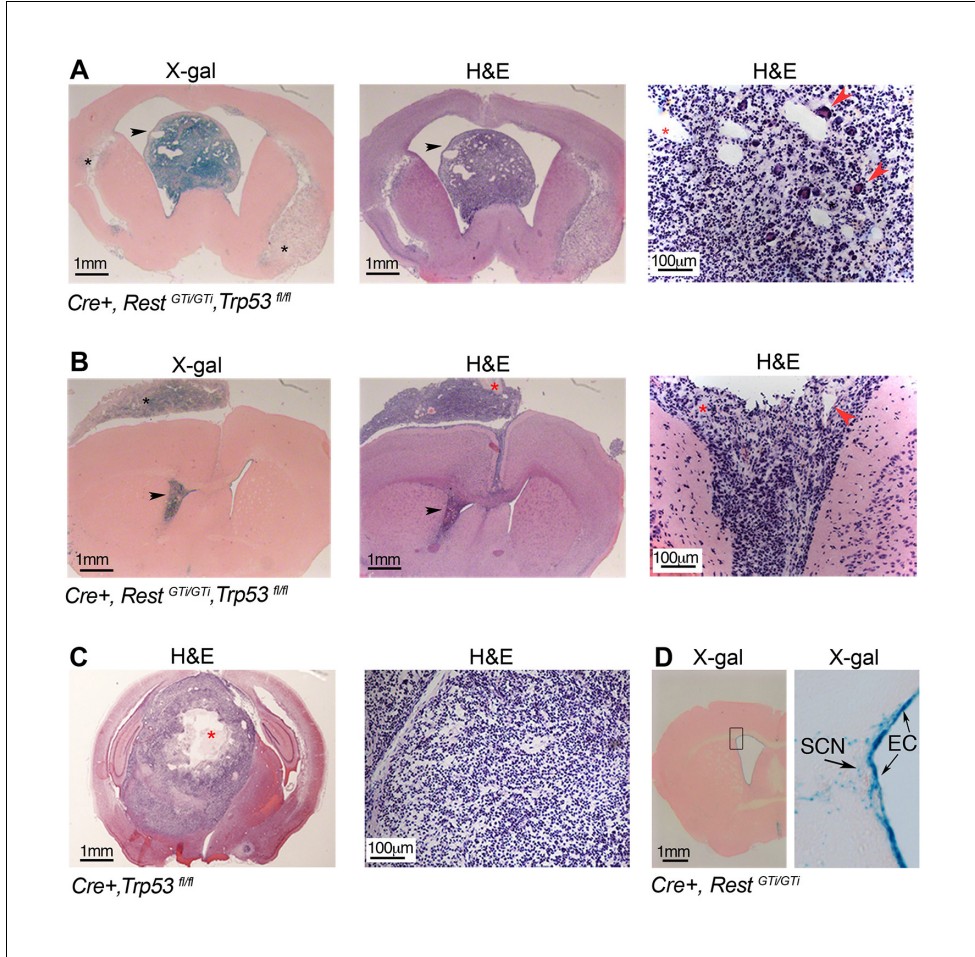

**Figure 7.** Persistent DNA damage due to loss of REST, in the context of loss of p53, promotes GBMs in adult. (**A**) and (**B**). Left panels, X gal staining of sections from *Cre+, Rest*$^{GTi/GTi}$ *Trp53*$^{fl/fl}$ mice at 7.5 months (**A**) and 10.5 months (**B**) of age. Arrowheads point to intraventricular tumors. Black asterisks indicate secondary tumor formation with subpial spread (**B**). Middle and right panels, H&E staining reveals abnormally increased cellular densities. Higher magnifications reveal atypical morphology of the densely packed cells, with little cytoplasm and multiple areas of new blood vessel formation (red arrowheads). Red asterisks indicate pseudopallisading necrosis. (**C**) Left panel, section from 11-month-old p53-deficient mouse brain showing a rare GBM, higher magnification image on right. Red asterisk, necrotic area. (**D**) Section from 9.5-month-old *Cre+, Rest*$^{GTi/GTi}$ mouse brain. Boxed area is enlarged at right. Arrows point to EC and SCN. EC, ependymal cells; GBM, glioblastoma; H&E, hematoxylin and eosin; SCN, stem cell niche.

The following figure supplements are available for figure 7:

**Figure supplement 1.** Immunostaining analysis defining grade IV (GBM) gliomas according to WHO classification.

**Figure supplement 2.** GBM from *Cre+, Rest*$^{GTi/GTi}$, *Trp53*$^{fl/fl}$ mice show high proliferation indices and expression of progenitor and NSC molecular markers.

## Adult Cre+, Rest$^{GTi/GTi}$, Trp53$^{fl/fl}$ mice develop proneural type GBM with primitive neuroepithelial tumor (PNET) characteristics

Previous studies support the idea that REST is a tumor suppressor outside of the nervous system (*Wagoner et al., 2010*; *Baye and Link, 2007*; *Wagoner et al., 2010*; *Westbrook et al., 2008*; *Gurrola-Diaz et al., 2003*; *Kreisler et al., 2010*; *Moss et al., 2009*), but tumors generally reflect the loss of more than one tumor suppressor. To test whether the DNA damage due to loss of REST, in the absence of the additional tumor suppressor p53, was persistent and would promote tumorigenesis, we maintained *Cre+, Rest$^{GTi/GTi}$, Trp53$^{fl/fl}$* mice into adulthood. We found 66% of 131 mice developed brain tumors at 9–11 months of age (*Figure 7A and B* and *Table 1*), and, of these, 48% were high-grade GBMs (*Louis et al., 2007*), based on proliferation indices, presence of mitotic figures, neovascularization with proliferating endothelial cells, and presence of necrosis (*Figure 7B*, *Figure 7—figure supplement 1*). Interestingly, tumors often consisted of undifferentiated cells with small cytoplasm (*Figure 7A and B*), characteristic of GBM with a primitive neuroectodermal-like component (PNET) (*Kouyialis et al., 2005*; *Shingu et al., 2005*; *Ohba et al., 2008*; *Song et al., 2011*). While common in the pediatric population, GBMs with PNET component are very rare in adults and are often thought to arise from clonal expansion of progenitor/tumor cells in vascular rich areas (*Perry et al., 2009*; *Zindy et al., 2007*). Anatomically, the GBMs we observed were often (36%) intraventricular or periventricular (*Figure 7A and B*), consistent with the location of progenitor/stem cells in the adult brain and REST expression in the neural stem cell (NSC) niche. In addition, the majority of tumors in the REST, p53-deficient mice were β-gal+ (*Figure 7A and B*), and expressed neural progenitor/stem cell markers (*Figure 7—figure supplement 2*), suggesting a progenitor/NSC tumor origin and consistent with the PNET character of GBM.

Similar to a previous report (*Zheng et al., 2008*), gliomas in the *Cre+, Trp53$^{fl/fl}$* mice were much less frequent (13/68) and only three of the 13 could be characterized as GBM (*Table 1*). In addition, none of the tumors were intraventricular or showed spread along the hemispheres (*Figure 7C*). No tumors formed in *Rest$^{GTi/GTi}$* mice and βgal+ cells were confined to ependymal and stem cells niche areas in SVZ (*Figure 7D*). The lack of tumors in control mice indicated that DNA damage due to loss of REST was insufficient by itself to cause tumors and/or that most of the damaged cells were eliminated by p53-mediated apoptosis.

Despite their heterogeneity, GBMs can be characterized based on their molecular expression profile into classical, mesenchymal, proneural, and neural subtypes correlated with certain mutations, chromosomal aberrations, severity and responses to therapy (*Verhaak et al., 2010*). We performed qRT-PCR analysis for a subset of the signature genes on 17 *Cre+, Rest$^{GTi/GTi}$, Trp53$^{fl/fl}$*tumors (*Verhaak et al., 2010*) (*Figure 8A*). The analysis indicated consistently high expression of Olig2, Erbb3, Ng2, Pdgfrα, and Nkx2.2 genes in GBM, independent of the manifestation of clinical symptoms or anatomical location of the tumor (*Figure 8A* and data not shown). The genes represented a typical constellation of a proneural GBM subtype (*Verhaak et al., 2010*) and were not up-regulated in normal cortex of control mice. The Olig2, Pdgfrα, Ebrb3, Ng2 and Nkx2.2 signature was maintained in tumor cells propagated in culture from *Cre+, Rest$^{GTi/GTi}$, Trp53$^{fl/fl}$*mice (data not shown). The expression of these genes also correlated with an oligodendrocyte-specific molecular signature identifying proneural type of GBM (*Verhaak et al., 2010*), and E13.5 OLIG2+ SOX2+ progenitors in brains from *Cre+, Rest$^{GTi/GTi}$, Trp53$^{fl/fl}$* mice exhibited DNA damage (*Figure 8B*).

## Conditional Rest KO mice produced by deleting Rest exon 2 exhibit normal brain size and lack apoptosis

Our findings on the *Cre+, Rest$^{GTi/GTi}$* mice were not recapitulated when we tested a brain-specific *Rest* KO with excision of exon 2 (*Cre+, Rest$^{fl/fl}$* mice) (*Gao et al., 2011*). In this line, the remaining exons still have the potential to encode several zinc finger domains in the DNA-binding domain as well as the nuclear localization signal (*Shimojo, 2001*) and C-terminal repressor domain (*Figure 9A*) (*Tapia-Ramirez et al., 1997*; *Thiel et al., 1998*). The average brain sizes of P45 *Cre+, Rest$^{fl/fl}$* females (0.46 SD 0.02, n=7) were not statistically different from the sizes of *Rest$^{fl/fl}$* littermate females (0.44 SD 0.02, n=9) or control *Rest$^{GTi/GTi}$* mice (0.46 SD 0.01, n=15). A potential explanation was provided by analysis of RNA and REST protein levels in E13.5 brain or progenitors from the *Cre+, Rest$^{fl/fl}$* mice. Exons 3 and 4 in the *Cre+, Rest$^{fl/fl}$* mice were transcribed (*Figure 9B*) and encoded an ~130 kDa C terminal REST peptide (hereafter REST$^C$) (*Figure 9C*) that was not present in the *Cre+,*

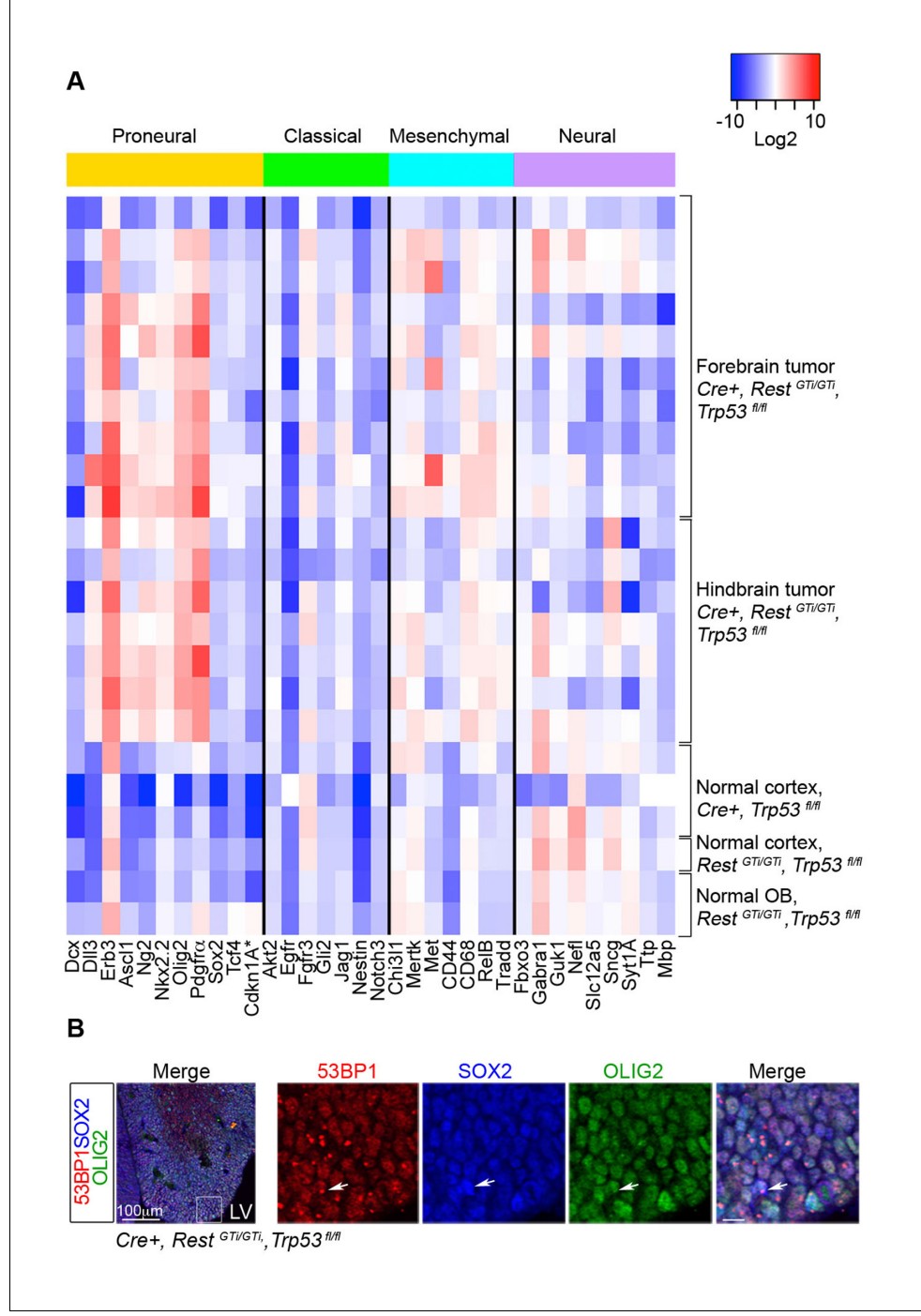

**Figure 8.** RNA analysis points to proneural type tumors. (**A**) Heat map representing qRT-PCR analysis of mRNA levels, normalized to 18S, of selected genes enriched in human proneural, classical, mesenchymal, and neural subtypes of GBMs. The values are color coded and plotted on Log$_2$ scale. (**B**) Representative immuno-labeling of LGE from E13.5 *Cre+, Rest$^{GTi/GTi}$, Trp53$^{fl/fl}$* mice showing 53BP1 foci in progenitor cells. Boxed area on the left image is enlarged on right images (scale bar, 10µm). Arrow points to cell with foci. GBM, glioblastoma; LGE, lateral ganglionic eminence; LV, lateral ventricle; qRT-PCR, quantitative real-time polymerase chain reaction.

*Rest$^{GTi/GTi}$* mice but remained in mouse embryonic stem cells ESCs) created using a similar deletion strategy (*Jorgensen et al., 2009*). The size of REST$^C$ suggests that an initiator methionine was utilized in exon 3.

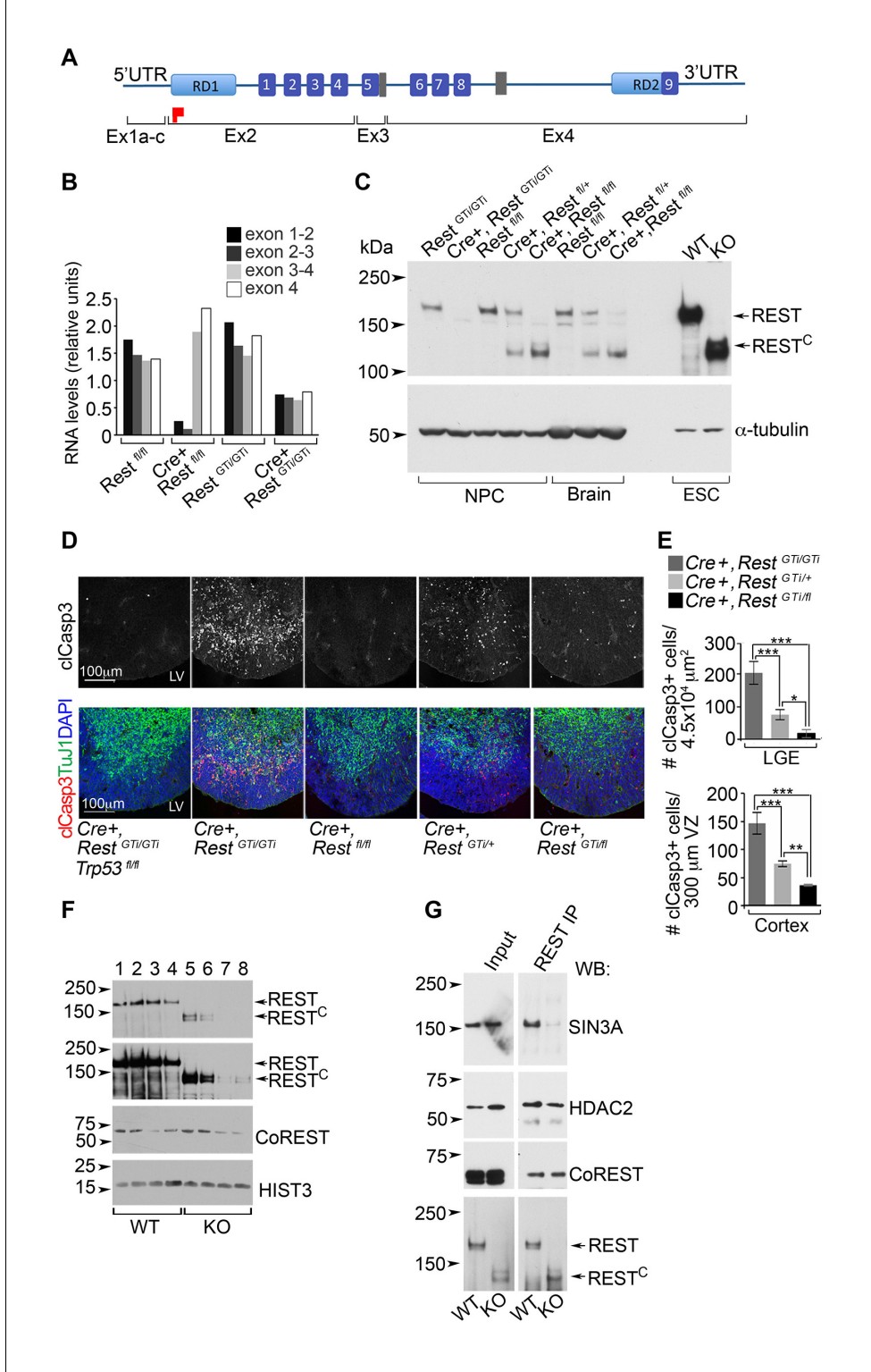

**Figure 9.** A C-terminal REST peptide, translated in conventional *Rest* KO mice, recruits co-repressors and protects against apoptosis. (**A**) Schematic of REST functional domains (top) and exons (Ex, bottom). (**B**) qRT-PCR analysis, relative to 18S RNA, spanning indicated exons from E13.5 brain. (**C**) Western blot analysis of neurospheres (NPC), E13.5 brain, and ESC lysates in indicated genotypes using antibody raised against the C-terminus of REST[6]. Full length REST protein migrates at ~200kDa[6]; 130 kDa peptide represents peptide containing the REST[C]. α-tubulin, loading control. (**D**) Representative immunostained images in cortical sections from indicated genotypes. (**E**)

*Figure 9. continued on next page*

*Figure 9. Continued*

Quantification of apoptotic cells, n=5–6 mice/genotype, 6–10 sections/mouse. Mean values and 95% CI are shown. Statistical significance was determined using Kruskal ANOVA with Dunn posthoc, (**F**) Representative Western blot from ESC nuclear extracts, n=3 independent experiments. Chromatin extracts were performed with increasing NaCl concentrations; 0 (lanes 1, 5); 100 mM (lanes 2, 6); 200 mM (lanes 3, 7); 500 mM (lanes 4 and 8). Top two panels represent different exposures of the same Western blot. (**G**) Representative Western blot analyses of REST and REST$^C$ immuno-complexes from ESC nuclear extracts. Extracts from ESC were immunoprecipitated with REST antibody and probed with antibodies indicated at right, n=3 independent experiments. *p< 0.05; **p<0.01; ***p< 0.001. RD, repressor domain; gray rectangle, nuclear localization signal; purple box, zinc finger motif; red flag, initiator methionine. ANOVA, analysis of variance; ESC, embryonic stem cell; qRT-PCR; REST$^C$, C terminal peptide of REST.

Because we had shown that apoptosis contributes to smaller brain size in the *Cre+, Rest$^{GTi/GTi}$* mice, the above results suggested that the normal brain size in *Cre+, Rest$^{fl/fl}$* mice was due to REST$^C$ activity that protected against apoptosis. To test this idea, we took advantage of *Cre+, Rest$^{GTi/+}$* mice that are hypomorphic with respect to brain size and apoptosis. These mice exhibit levels of apoptosis 50.1 and 37.4% lower than *Cre+, Rest$^{GTi/GTi}$* mice depending on the brain region (Cortex: *Cre+, Rest$^{GTi/GTi}$*, 146.9, CI 127.6–127.6; *Cre+, Rest$^{GTi/+}$* 74.69, CI 69.6 79.8; LGE: *Cre+, Rest$^{GTi/GTi}$*, 203.9, CI 169.1–238.6, *Cre+, Rest$^{GTi/+}$*, 76.3, CI 60.8–91.8) (*Figure 9D and 9E*). This result is also reflected in fewer and less aggressive gliomas in the context of p53 deletion in *Cre+, Rest$^{GT/+}$*, *Trp53$^{fl/fl}$* mice (*Table 1*).

To test whether REST$^C$ could block apoptosis in neural progenitors, we generated *Cre+* mice heterozygous for the *Rest$^{GTi}$* and *Rest$^{fl}$* alleles (*Cre+, Rest$^{GTi/fl}$*), and compared clCasp3 labeling with that of *Cre+, Rest$^{GTi/GTi}$* mice. The results indicated significantly less apoptosis in the *Cre+, Rest$^{GTi/fl}$* mice in both cortex and LGE, with decreases to 24.8 and 9.1% of *Cre+ Rest$^{GTi/GTi}$* values, respectively (Cortex: *Cre+, Rest$^{GTi/fl}$*, 36.4, CI 34.9–37.9; LGE: *Cre+, Rest$^{GTi/fl}$* 18.54, CI 6.3–30.8) (*Figure 9D and E*). This result demonstrates that the remaining REST$^C$ protein generated using Exon 2 deletion can still function in progenitors to prevent the DNA damage and apoptosis that results from complete loss of REST.

## REST$^C$ can still recruit co-repressors to chromatin

Previous studies have demonstrated that both C- and N-terminal domains of REST exhibit repressor activity through recruitment of distinct co-repressor complexes (*Andres et al., 1999*; *Ballas et al., 2001*; *Grimes, 2000*; *Dingledine et al., 1999*; *Naruse et al., 1999*; *Roopra et al., 2000*). Thus, the REST C-terminal peptide could prevent DNA damage and apoptosis by maintaining its known function and recruiting co-repressors directly to the chromatin. Indeed, REST$^C$ was bound to chromatin prepared from ESCs, albeit not as effectively as full length REST (*Figure 9F*). REST$^C$ was also in immuno-complexes with the known REST co-repressors HDAC2 and CoREST (*Andres et al., 1999*; *Ballas et al., 2001*) but not with the co-repressor SIN3A (*Figure 9G*), which is recruited only by the N-terminus of REST (*Grimes, 2000*;*Dingledine et al., 1999*; *Naruse et al., 1999*; *Roopra et al., 2000*) .

## Discussion

Here, we show a new role for REST in protecting genomic integrity in cycling neural progenitors. Premature loss of REST results in prominent DNA damage during S phase, as well as in chromosomal abnormalities, resulting ultimately in apoptosis of neurons that normally mature during early to intermediate neurogenesis. These events are distinct from the robust up-regulation of neuronal genes that are predicted prima facie from premature loss of REST, and point to a mechanism for ensuring that proper terminal neuronal differentiation occurs only after exit from the cell cycle.

*The S phase of the cell cycle in neural progenitors is a critical decision point for proper terminal neuronal differentiation.* REST is an ideal candidate for probing links between proliferation and terminal neuronal differentiation because it represses a large coterie of genes that are required for mature neuronal functions. As a repressor of these genes, REST is down-regulated dramatically during embryonic neurogenesis to allow elaboration of the terminally differentiated phenotype.

However, REST is still associated with its chromatin modifiers on neuronal gene chromatin in the dividing progenitors (*Figure 2F* [*Ballas et al., 2005*]). The precise timing of release of REST from chromatin during the cell cycle has never been determined. Our results indicate that its normal removal must coincide with completion of cell division, because loss of REST and its co-repressors in cycling progenitors results in DNA damage, early cell cycle exit, and eventual apoptosis.

In particular, our results suggest strongly that premature loss of REST causes DNA damage during S phase. We find significantly more mutant than control cells having a DNA damage response and a larger percentage of BrdU+ cells in the VZ with damage foci compared to BrdU− cells (*Figure 6G and H*). Both DNA damage and premature cell cycle exit persist in the *Cre+, Rest$^{GTiGTi}$, Trp53$^{fl/fl}$* mice, indicating that these events can occur independently of activation of the apoptotic pathway. In most cases, apoptosis is a common outcome of DNA damage (*Houlihan and Feng, 2014*). Because a majority of the MAP2+ cells that accumulated at the VZ/SVZ boundary in the mutant were also positive for clCasp3, we were unable to determine whether expression of MAP2 followed a normal differentiation pathway or an abnormal up regulation of MAP2 contributed to the apoptosis. Time lapse imaging of individual cells in the mutant may answer this question in the future.

Even with apoptosis and smaller brain size, *Cre+, Rest$^{GTi/GTi}$* mice lived to maturity. This situation contrasts with the widespread apoptosis and loss of entire brain structures that is often characteristic of loss of other chromatin complexes during embryogenesis. For example, brain KO of both HDACs 1 and 2 results in embryonic lethality with evidence of apoptosis throughout the brain (*Hagelkruys et al., 2014*; *Montgomery et al., 2009*). Similarly, brain deletion of TOPBP1, which has a global role in maintaining genomic stability, causes widespread apoptosis and loss of entire brain structures (*Lee et al., 2012*). The more robust phenotype of these factors is likely due to their inclusion in many different transcriptional complexes, whereas REST binding sites are restricted to a more limited set of sites in the genome.

The brain phenotype of our mice was most similar to loss of the nuclear factor NDE1 reported recently to safeguard genomic integrity during S phase through interactions with the cohesin complex, implicated in replication fork fidelity (*Houlihan and Feng, 2014*). We did not identify NDE1 in any of our purified REST complexes. Furthermore, unlike the nearly complete brain size rescue in the NDE1, p53 compound KO mice, the deletion of *Trp53* in *Rest$^{GT}$* mice, although it did block apoptosis, allowed only partial recovery to normal brain size, indicating residual effects of REST loss that did not occur with loss of NDE1(*Houlihan and Feng, 2014*). These distinctions suggest different underlying mechanisms for their functions in neural progenitors. Interestingly, significant up-regulation of neuronal genes during development does occur outside the nervous system in REST-deficient mice (*Figure 1F* and *Aoki et al., 2012*; *Paquette et al., 2000*), indicating distinct mechanisms for S phase surveillance in non-neural and neural tissues.

*Premature loss of REST from neural progenitors, coupled with loss of the tumor suppressor p53, leads to invasive GBM.* The DNA damage due to loss of REST, in the absence of apoptosis with p53 deletion, persisted into adulthood and led to primarily proneural type GBM with PNET character (*Kouyialis et al., 2005*; *Shingu et al., 2005*; *Ohba et al., 2008*; *Song et al., 2011*). However, some tumors also expressed markers typical for mesenchymal and neuronal GBM types (*Figure 8A*, *Figure 7—figure supplement 2*), reflecting the fact that GBM are very heterogeneous, even when derived from single clones from patients (*Yung et al., 1982*; *Wikstrand et al., 1983*) including heterogeneity in REST expression (*Wagoner and Roopra, 2012*; *Kamal et al., 2012*; *Conti et al., 2012*). While the *Rest* gene was transcriptionally inactive due to the GT, the *Rest* promoter was active, evidenced by β-gal staining, in the tumor cells in *Cre+, Rest$^{GTi/GTi}$, Trp53$^{fl/fl}$* mice (*Figure 7A and B*), likely reflecting their progenitor/NSC character. Up-regulation of *Rest* mRNA and protein are a prominent feature in some human GBMs and medulloblastomas (*Wagoner and Roopra, 2012*; *Kamal et al., 2012*; *Conti et al., 2012*; *Su et al., 2006*; *Majumder et al., 2000*). Only a small percentage (23%, 3/13 gliomas) of *Cre+, Trp53$^{fl/fl}$* mice developed GBM with PNET pathology, and these lacked subpial spread. Taken together, our results reinforce the idea, proposed previously (*Wagoner et al., 2010*; *Westbrook et al., 2008*; *Gurrola-Diaz et al., 2003*; *Kreisler et al., 2010*; *Moss et al., 2009*) for non-neuronal cells, that DNA damage due to loss of REST does not initiate tumors but rather promotes transformation and migration.

*A C-terminal repressor domain in REST protects genomic integrity in neural progenitors.* Because REST recruits chromatin modifiers to repress its target genes, it seems reasonable to propose that

DNA damage effects from premature loss of REST reflect a failure to properly re-establish correct chromatin modifications during the cell cycle. There are two well-established repressor domains in the N and C termini of REST (*Tapia-Ramirez et al., 1997*; *Thiel et al., 1998*). Each of these domains recruits histone deacetylases, as well as enzymes that methylate or demethylate histones (*Battaglioli et al., 2002*; *Mulligan et al., 2008*; *Lunyak, 2002*; *Lee et al., 2005*). We showed that in a conventionally targeted REST KO mouse line targeting *Rest* exon2, remaining C terminal exon(s) are translated into a protein that can bind neuronal chromatin and recruit the CoREST/HDAC complex. The presence of REST$^C$ in the *Rest* GT heterozygote greatly reduced DNA damage and apoptosis, consistent with the idea that proper chromatin modifications are required to pass the S phase surveillance test. A study using a different *Rest* KO model, which targets exon 4 and leaves intact sequences coding for N-terminal repressor exons (*Aoki et al., 2012*), did not report microcephaly or DNA damage. It is thus possible that the remaining N-terminal peptides are functional, consistent with the ability of both N- and C-terminal repression domains of REST to mediate repression function (*Tapia-Ramirez et al., 1997*). It will be of interest to determine DNA damage and premature differentiation are separable events, and this may be resolved through more in-depth comparisons between *Rest* GT and conventional *Rest* KO mouse models.

*A new model for REST function during embryonic neurogene*sis. We propose that REST repression protects the integrity of neuronal genes whose expression must be delayed until terminal differentiation. There is no evidence for direct interactions of REST with DNA replication machinery. Therefore, we suggest that REST protection occurs by maintaining proper chromatin modifications during S phase. This could be achieved simply by preventing premature expression of terminal neuronal genes, thereby preventing aberrant chromatin modifications and premature cell cycle exit, and/or by functions of components in the REST repressor complex dedicated to DNA damage control. Although the magnitude and number of mRNAs de-repressed in the REST mutant at E12.5 was very low (*Supplementary file 2*), it is possible that abnormally timed transition of their chromatin from repressor to activator marks could contribute to a DNA damage response. With respect to the REST complex, several associated factors have the potential to protect against DNA damage. Intriguingly, FACT (FAcilitates Chromatin Transcription) turned up recently as a member of the REST complex in ESCs (*Mcgann et al., 2014*) In mammalian cells and yeast, loss of FACT results in DNA double-strand breaks that interfere with DNA replication through the formation of R loops (*Herrera-Moyano et al., 2014*).

Further studies are required to test this new model. However, our current results indicate that a functional REST repressor complex is required for proper cell cycle transition during neurogenesis, and that normal loss of REST repression from neuronal gene chromatin is timed precisely to coincide with cell cycle exit and the 'permission' to terminally differentiate.

# Materials and methods

## Experimental procedures

### Mouse strains

*Rest* $^{GT(D047E11)}$ (*Rest$^{GT}$*) mutants were established by blastocyst injection of the D047E11 GT clone (GenBank Acc.: DU821609). *Rest$^{GTi(D047E11)}$* mice carrying the inverted GT vector (*Rest$^{GTi}$*) were obtained by crossing to Flpe deleter mice (*Dymecki et al., 2000*). GT vector orientation was determined as described (*Schnutgen et al., 2005*). Mouse strain *Rest$^{fl/fl}$* was described previously (*Gao et al., 2011*). Mice carrying *Trp53$^{fl}$* allele, *Nestin Cre* and *hGFAP Cre* transgenes were from Jackson Laboratories (Bar Harbor, ME), stocks # 008462, 003771, and 004600, respectively, Strains were backcrossed to C57BL/6J (*Nestin Cre*, *Rest$^{GTi}$* and *Trp53* $^{fl}$ lines) and to FVB background (*hGFAP Cre line*) for at least 10 generations. Mice were genotyped for *Rest$^{GTi}$*, *Rest$^{GT}$*, or *Rest$^+$* alleles using the following primers: GTA5: 5'-tggatgttgaggtccgttgtg-3', GTB5: 5'-ggctacg-gatcccttcttccc-3'and GTB1: 5'aacggcccccgacgtccctgg-3' to reveal 480 bp product (wild type, GTB5/GTB1 amplicon) and 600 bp (mutant, GTA5/GTB1 amplicon). *Cre* transgene was validated using primers CreA1: 5'-tgctgtttcactggttatgcg-3' and CreB1: 5' ttgcccctgtttcactatcca-3'. Genotyping for *p53$^{fl}$*allele was according to Jackson Laboratories.

## Immunohistochemistry

Embryonic heads (E13.5 or younger), dissected brains (E14.5-P1), or transcardially perfused brains from adult animals, were fixed in 4% formalin in phosphate-buffered saline (PBS) at 4°C overnight, equilibrated in 30% sucrose and embedded in TFM tissue frozen medium (TBS) for frozen sectioning. Age- and position- matched sections were dried, post-fixed with cold acetone and stained using standard immunohistochemical techniques. Antigen retrieval in 10 mM sodium citrate, pH 6.0, 0.1% Tween-20, 95°C, 10 m, or proteolytic permeabilization with 0.1% Trypsin/0.1% CaCl2 in 10 mM Tris, pH 7.5 for 5 m at room temperature were used to reveal nuclear proteins. Standard 30 m 2 M HCl treatment at RT, followed by neutralization in 0.1 M Borate was used for BrdU immunostaining. Sections were stained with secondary species-specific antibodies conjugated to Alexa-488, Alexa-555, or Alexa-647 (Invitrogen, Waltham, MA), and counterstained with DAPI to reveal nuclei. Immunostaining was analyzed with a confocal fluorescent microscope (Zeiss LSM710 Axiovert). Images were acquired using Zeiss LSM5 (Carl Zeiss, Oberkochen, Germany) and processed using ImageJ and Photoshop CS3 (Adobe, San Jose, CA).

## Commercial antibodies

Rabbit anti-CoREST[67], (1:1000); rabbit anti-cleaved Caspase 3 (Cell Signalling, Davers, MA, 1:300); rabbit anti-Ki-67 (clone MM1, Novocastra, Newcaste, UK, 1:100) rabbit anti-DCX (Cell Signaling, 1:300); rabbit anti-phospho histone H2A.X (Ser139)(Cell Signalling, 1:100 for IF and 1:1000 for WB); Histone 3 (Cell Signalling, 1:1000); mouse anti-TuJ1 (Covance, Princeton, NJ, 1:500); rabbit anti-PAX6 (Covance, 1:300); rabbit anti-SOX2 (Millipore, Billerica, MA, 1:300); mouse anti-MAP2 (Millipore, 1:300); rabbit anti-TBR1 (Millipore, 1:500); mouse anti-GFAP (Millipore, 1:500); mouse anti-SOX2 (RD Systems, Minneapolis, MN, 1:100); rat anti-CTIP2 (Abcam, Cambridge, MA, 1:300); rabbit anti-TBR2 (Abcam, 1:300); mouse anti-SATB2 (Santa Cruz Biotechnology, Santa Cruz, CA, 1:100); rabbit anti-Olig2 (Millipore, 1:300); rabbit anti-mSin3 (Santa Cruz Biotechnology, 1:1000); mouse anti-phospho ATM (Ser 1981) (ECM Biosciences, Versailles, KY, 1:1000 for WB and 1:300 for IF); mouse anti-ATM (Sigma-Aldrich, St Louis, MO, 1:1000); rabbit anti-HDAC2 (Life Technologies, Waltham, MA, 1:5000); rabbit anti-p53BP1(Calbiochem, Billerica, MA, 1:500), cleaved Caspase3-conjugated to Alexa-647 (Cell Signalling, 1:50). Mouse anti-α-tubulin (clone AA4.3, 1:5000), mouse anti-actin (clone JLA20, 1:1000), anti-RC2 (1:100) and mouse anti-BrdU (clone G3G4, 1:200), were obtained from the Developmental Studies Hybridoma Bank developed under the auspices of the NICHD and maintained by the University of Iowa, Department of Biology.

## REST antibodies

Rat monoclonal anti-mouse REST antibody 4A9[91] was used (1:300) in *Figure 1I*. Rabbit anti-human REST antibody REST-C[7], was used in *Figure 1D, 2D, 9C, F and G* (1:1000), and in *Figure 2G* (1:100). The rabbit anti-mouse REST antibody RESTJ used in *Figure 9G* was raised against amino acids 889–1035 of REST (within *Rest* exon 4) after expression in bacteria (BL2, DE3, Agilent, Santa Clara, CA) followed by glutathione-agarose affinity purification. Purified peptide was used for immunizations of rabbits and collection of antiserum (Covance). Serum was further passed over GST-agarose (Thermo Scientific, Waltham, MA) to deplete of anti-GST epitopes and affinity purified against the GST-REST peptide antigen coupled to glutathione-agarose (Thermo Scientific). Specificity of this antibody was confirmed by Western blotting (1:1000) of extracts from a mouse REST KO[66] and WT ESC cells (data not shown).

## Western blot

Proteins were lysed in cold lysis buffer containing 50 mM Tris, pH 7.5, 150 mM NaCl, 1 mM ethylenediaminetetraacetic acid (EDTA), 0.1 mM sodium vanadate, 10 mM β-glycerophosphate; 10 mM NaF, protease inhibitors (Roche Basel, Switzerland), 1% Triton X-100 and 10% glycerol and resolved on 3–8% pre-cast gradient Tris-Acetate gels (Invitrogen, Waltham, MA). Blotting with primary antibodies was carried out at 4°C overnight followed by incubation with the appropriate secondary antibodies and developed using chemiluminescent West Pico detection kit (Thermo Scientific).

## Chromatin immunoprecipitation

Analyses were carried out as described (*Ballas et al., 2001*.) Briefly, 200 µg of protein lysate from embryonic brain or cultured neurospheres with cross-linked and sheared chromatin were immuno-precipitated overnight at 4°C using rabbit anti-REST antibodies generated against the C-terminus (*Ballas et al., 2005*). Normal rabbit IgG was used as immunoprecipitation negative control. After reversal of cross-links, DNA and 10% input samples were purified using Qiagen (Valencia, CA) columns. For qRT-PCR analyses, primers were designed within 300–500 bp of the RE1 site and ChIP DNA amounts were determined from standard curve and normalized to the input DNA. PCR primer sequences used for qRT-PCR in *Supplementary file 3*.

## Real-time quantitative PCR (qRT-PCR)

RNA was extracted using Trizol (Invitrogen) and treated with DNAse (Ambion, Waltham, MA). Reverse transcription reactions were performed using 1 mg-500 ng of total RNA with Superscript III (Invitrogen). qPCR was performed in an Applied Biosystems PRISM 7900HT Fast Real Time PCR system with SYBR green PCR master mix (Applied Biosystems, Waltham, MA) using the same cycling conditions. Relative abundance of each cDNA was determined according to the standard curve and normalized to 18S RNA levels. Samples were diluted 1:50 for analysis of 18S RNA levels and 1:5 for analysis of test genes.

## In situ hybridization analysis

Antisense and sense probes for REST mRNA detection were as described (*Chong et al., 1995*). Plasmids containing probes were linearized and transcribed using T7 RNA polymerase (Promega, Madison, WI) and DIG-labeling methodology (Roche). In situ hybridization analyses were carried out on frozen brain sections as described (*Sciavolino et al., 1997*).

## Cell culture

Neurospheres were isolated from E13.5 cortices and cultured (*Reynolds and Weiss, 1992*) in neurobasal media (Invitrogen) supplemented with 2 mM L-glutamine, 100 U/ml penicillin, 100 mg/ml streptomycin, B27 supplement, 20 ng/ml epidermal growth factor (Invitrogen), and 10 ng/ml fibroblast growth factor (Invitrogen). Neurospheres were passaged every 3–4 days using Accutase (Invitrogen). ESC were cultured as described (*Mcgann et al., 2014*).

## Chromatin association

Analysis was carried out as described (*Dykhuizen et al., 2013*). In brief, nuclei were isolated with buffer A (25 mM HEPES, pH 7.6, 5 mM MgCl$_2$, 25 mM KCl, 0.05 mM EDTA, 10% glycerol, 0.1% NP-40 and protease inhibitors (Roche)) and $1.5 \times 10^6$ nuclei/per condition were extracted for 20 min on ice in 20 mM Tris-HCl, pH 6.3, 3mM EDTA with 0, 100, 200, 300 and 500 mM NaCl concentration. Chromatin extracts were centrifuged for 20 min at 25K to isolate chromatin. The lysate was removed and chromatin pellet was re-suspended in sodium dodecyl sulfate (SDS) lysis buffer using sonication, followed by Western blotting for chromatin association.

## Co-immunoprecipitation

Nuclei were isolated in buffer A and extracted in IP buffer (150mM NaCl, 50 mM Tris-HCl, pH7.5, 0.5% TritonX-100, 0.5 mM EDTA, 10% glycerol with protease (Roche) and phosphatase inhibitors). DNA and RNA were digested using Benzonase (Sigma) for 30 min at 4°C, lysates pre-cleared with protein G Dynabeads (Life Technologies) and immunoprecipitated with constant rotation overnight at 4°C using 1 µg of primary rabbit anti-mouse REST antibody or rabbit IgG, followed by 2 hr of incubation with protein G Dynabeads. Immunoprecipitates were washed four times on ice using IP buffer with constant rotation, beads were re-suspended in SDS loading buffer and analyzed by Western blotting.

## LeX+ purification of progenitors

Telencephalons from E13.5 brain were mechanically dissociated and labeled with anti-LeX/SSEA-1 antibody conjugated to PE (BD Pharmingen, San Diego, CA) for 20 min in PBS containing 1%

bovine serum albumin and 1 mM EDTA on ice. Cells were washed and incubated with anti-PE magnetic microbeads (Miltenyi Biotech, Gladbach, Germany), for additional 20 min on ice, washed and applied to magnetic-activated cell sorting column on magnetic separator. Unbound cells were washed and bound cells eluted as suggested by the manufacturer (Miltenyi Biotec). Cells were collected by centrifugation and lysed in Trizol for RNA preparation.

### Statistical analysis

Statistical analyses, including t-tests, analysis of variance (ANOVA), calculation of means and 95% CI, were performed using Prism5 (Graphpad, La Jolla, CA). For the selection of the appropriate tests, data were analyzed for a normal distribution. For normally distributed samples, we used ANOVA followed by Tukey posthoc for comparisons of more than two groups or unpaired two-tailed t-test with Welch correction for two-group comparisons. For the non-normally distributed data sets, non-parametric Kuskal–Wallis ANOVA with Dunn posthoc test for more than two group comparisons and Mann–Whitney t-tests for two groups were used for comparisons. In addition, Prism5 software was used to plot means and 95% CI intervals or standard deviations on figures.

### Microarray analysis

Total RNA from E12.5 brains was extracted with Trizol (Invitrogen) and further purified using RNAeasy purification columns (Qiagen). 1µg of total RNA was used to generate cDNA libraries for hybridizations on Affymetrix (Santa Clara, CA) Gene Chip Array using Mouse 430_2 chip (*Supplemental file 1*, GEO acc.: GSE68459) and Ilumina Expression Ref8_2 BeadArray (*Supplemental file 2*, GEO acc.: GSE68368) at the OHSU Gene Profiling Shared Resource. Affymetrix microarray quality assessment and RMA normalization was carried out using AffyPLM package (Bioconductor, http://www.bioconductor.org). Illumina (San Diego, CA) BeadChips Data were processed with BeadStudio version 3.0 and the lumi package in Bioconductor, which was used for quality control and quantile normalization. Normalized data was log2 transformed and analyzed using GeneSifter (Perkin Elmer, Waltham, MA), using t-test to obtain raw p values between different genotypes and Benjamini-Hochberg false discovery rate for adjusted p values.

### Heat map generation

Heatmaps were generated using the Heatplus package in R (*Ploner et al., 2014*). Heatplus: Heatmaps with row and/or column covariates and colored clusters. R package version 2.12.0.

## Acknowledgments

The following grants supported this work: NIH grant NS22518 to GM; Bayerische Staatsministerium für Bildung, Wissenschaft und Kunst (Research Network 'Human Induced Pluripotent Stem cells' (ForIPS) and the European Commission (Health-FH-2010-242129, SyBoss) to WW, KNDD2 grant FKZ01GI1005D and Federal Ministry for Education and Research (BMBF) grant 'Funktionelle Tiermodelle für die Kandidatengene der Alzheimerschen Erkrankung' (01GS08133) to TF and WW; NIH grants R01AG032383, K02AG041815, and R21NS090926, the Welch Foundation I-1660, and Texas Institute for Brain Injury and Repair to JH. We thank Dr Sakir Humayun Gultekin for advice in characterization of tumor pathology. Dr. Rongkun Shen and the OHSU Gene Profiling Resource performed and helped analyze the microarray results, Glen Corson helped with antibody generation and characterization, and Jennifer Miller, Andrea Ansari and Michaela Voorhees performed mouse husbandry, genotyping, and histology.

## Additional information

### Funding

| Funder | Grant reference number | Author |
|--------|------------------------|--------|
| Howard Hughes Medical Institute | | Gail Mandel |
| National Institutes of Health | NS22518 | Gail Mandel |
| National Institutes of Health | R01AG032383 | Jenny Hsieh |

| Welch Foundation | I-1660 | Jenny Hsieh |
| Texas Institute for Brain and Injury Repair | | Jenny Hsieh |
| KNDD2 | FKZ 01 GI 1005D | Thomas Floss |
| Federal Ministry for Education and Research Grant | 01GS08133 | Thomas Floss |
| European Commission | Health-FH-2010-242129, SyBoss | Wolfgang Wurst |
| Bayerisches Staatsministerium für Bildung und Kultus, Wissenschaft und Kunst | Human Induced Pluripotent Stem cells | Wolfgang Wurst |
| National Institutes of Health | K02AG041815 | Jenny Hsieh |
| National Institutes of Health | R21NS090926 | Jenny Hsieh |

The funders had no role in study design, data collection and interpretation, or the decision to submit the work for publication.

## Author contributions

TN, Conception and design, Acquisition of data, Analysis and interpretation of data, Drafting or revising the article; JMcG, GM, Conception and design, Analysis and interpretation of data, Drafting or revising the article; KM, Acquisition of data, Analysis and interpretation of data, Drafting or revising the article; JH, Acquisition of data, Drafting or revising the article, Contributed unpublished essential data or reagents; WW, Drafting or revising the article, Contributed unpublished essential data or reagents; TF, Conception and design, Acquisition of data, Drafting or revising the article, Contributed unpublished essential data or reagents

## Ethics

Animal experimentation: This study was performed in strict accordance with the recommendations in the Guide for the Care and Use of Laboratory Animals of the National Institutes of Health. All of the animals were handled according to approved institutional animal care and use committee (IACUC) protocols (#IS0025) of the Oregon Health and Science University.

# Additional files

## Supplementary files

• Supplementary file 1. Genes significantly upregulated in E12.5 *Cre+, Rest*$^{GTi/GTi}$ brain vs Rest$^{GTi/GTi}$ brain revealed by microarray analyses.

• Supplementary file 2. Genes significantly upregulated in brains of E12.5 *Cre+, Rest*$^{GTi/GTi}$, *Trp53*$^{fl/fl}$ *mice vs Rest*$^{GTi/GTi}$, *Trp53*$^{fl/fl}$ revealed by microarray analyses.

• Supplementary file 3. Primer sequences used in the study.

## Major datasets

The following datasets were generated:

| Author(s) | Year | Dataset title | Dataset ID and/or URL | Database, license, and accessibility information |
| --- | --- | --- | --- | --- |
| Tamilla N, James M, Karin M, Shen R, Thomas F, Wolfgang W, Gail M | 2015 | Expression analyses of E12.5 embryonic brains from Nestin Cre+, Rest GTi/GTi vs Rest GTi/GTi litermates | http://www.ncbi.nlm.nih.gov/geo/query/acc.cgi?acc=GSE68459 | Publicly available at the NCBI Gene Expression Omnibus (Accession no: GSE68459). |

| Nechiporuk T, McGann J, Mullendorff K, Floss T, Wurst W, Mandel G | 2015 | Expression analyses of E12.5 embryonic brains from Nestin Cre+, Rest GTi/GTi, p53 fl/fl vs Rest GTi/GTi, p53 fl/fl littermates | http://www.ncbi.nlm.nih.gov/geo/query/acc.cgi?acc=GSE68368 | Publicly available at the NCBI Gene Expression Omnibus (Accession no: GSE68368). |

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
