## [Decision Letter]

Thank you for submitting your work entitled "The REST remodeling complex protects genomic integrity during embryonic neurogenesis" for peer review at *eLife*. Your submission has been favorably evaluated by Fiona Watt (Senior Editor) and two reviewers. One of the two reviewers, Michael Rosenfeld (Reviewer #1), has agreed to reveal his identity.

Below you will find two full reviews of your work. As you can see, the reviewers are highly positive about your findings and request only minor changes. You may want to include some discussion about the relevance of your findings to other tumour types and speculate about inducible REST deletion in adult brain. Please include the quantitation requested in Figure 3 and please consider including some additional markers in Figure 5.

*Reviewer #1:*

General assessment and major comments:

In the manuscript titled "The REST remodeling complex protects genomic integrity during neurogenesis", the authors reported a very surprising and interesting finding that a previously uncharacterized genome protective role of REST remodeling complex during S-phase of cell cycle in embryonic neurogenesis, which may be the underlying mechanism of a tumor suppressor role of REST in cancer progression and a neuron protective role of REST in age-associated neurodegeneration.

Although REST is a critical transcriptional regulator in neurogenesis, the lack of an obvious neuronal phenotype in the REST knockout model prompted the author "re-open" the case by generating a novel mouse model of REST deficiency. Using a conditional gene-trap strategy, the authors discovered a novel role of REST in genome protection during neurogenesis. Further, they provide in vivo evidence of REST in tumorigenesis using REST and p53 compound knockout mice. In the end, they compared the knockout strategies of two REST mouse models and pinpointed the critical difference between the knockout mouse design strategies that leads to "no obvious neuronal phenotype" conclusion based on the previous REST knockout model.

The role of REST in genome protection during S-phase is a rather surprising finding. The authors revealed an increased number of neurons with positive stains of DNA damage marks such as rH2AX, pATM and p53BP1 in REST KO mice. In addition, they found an increased number of abnormal chromosomal bridges in the REST KO neurons, indicating an important role of REST in S-phase dynamic during neurogenesis. It will be an interesting topic in the future as if REST is involved in DNA damage repair during cancer progression.

There are many links between REST and tumorigenesis previously. However, the causal role of REST deficiency in cancer progression is not yet clearly demonstrated. The authors reported that a significant increase of tumors observed in REST and p53 compound knockout mice, indicating a causal role of REST in tumorigenesis. The author used Nestin Cre in this study; it will be interesting in the future to determine if deletion of REST in the adult brain using inducible Cre on the p53-null background can increase tumorigenesis. In addition, it will be an interesting topic in the future that the tumor suppressor role of REST can be extended to other tumors such as breast cancer observed in human patients.

Mouse models can provide invaluable information in biomedical research. However, it is really rare to "re-open" the investigation if there is a mouse model available since it is considered risky to regenerate a different mouse model with a hope of a different result. The authors' efforts paid off in the end, indicating that micro differences in the experimental design can have quite different outcome. Therefore, a better lesson learned here is to pay more attention to the details of experimental design if the actual results didn't match the predicted based on the hypothesis.

Minor comments:

Overall, the manuscript is well written except a few minor points to be clarified during revision.

1) The authors performed REST ChIP on a few gene targets (Figure 2). Where are the REST binding sites, on promoters or RE-1 elements of *Glrp* or *Snap25*? In addition, what are the expression levels of *Glrp* or *Snap25* in control and REST KO brains?

2) Can the authors clarify CTIP2+ cells (Figure 3) between control and REST KO brains? It appears that CTIP2+ cells are increased (Figure 3), but there are not significant changes reported (Figure 3).

3) What are the percentage of clCasp3/BrdU + or 53BP1/BrdU + cells in control mice (Figure 6)?

4) Recently, a few non-coding RNAs or splicing variants have been identified in REST locus from many human cancer cells. Therefore, it is possible that REST locus encodes both REST protein and ncRNAs with distinct biological functions. If the authors can determine the expression of non-coding RNAs or splicing variants in both REST KO strategies, it may help the understanding of the biological roles of REST and the non-coding RNAs in REST locus.

Additional data files and statistical comments:

Can the authors provide transcriptional profiling data comparing control with REST KO brains discussed in the manuscript?

Reviewer #2:

General assessment and major comments:

In this study Nechiporuk et al., uncover an unexpected role for REST in maintaining genomic integrity. They work with a complete loss of function mouse strain (a gene trap reporter) and find that conditional deletion in neural progenitors results in a distinct phenotype from that reported previously. They observe microcephaly in the conditionally deleted mice, and this is consistent with the recent links to human microcephalies. The REST null neural progenitors are also susceptible to malignant transformation when p53 is concomitantly removed. The discrepant findings with the previous knockouts are reconciled by their demonstration that the exon 1 encoded C terminal peptide remains expressed and functional; past mice were therefore not null.

These are interesting and important observations that will be of broad interest to several different research communities. The major novelty is in the realization that REST may protect genomic integrity during cell division.

The REST GT mice have an embryonic lethal phenotype consistent with the previously reported knockout and characterization of this mouse is performed well. They then create mice with a genetic inversion of the GT and breed these to a Nestin Cre driver to achieve conditional deletion. These data are convincing and point to a reduced overall brain size and a thinner cerebral cortex. They go to show that this is due to reduced numbers of cycling cells and increases in apoptotic cells. Figure 5 is particularly convincing and the compelling observation indicating cell death rather than a promiscuous premature neuronal differentiation.

Of particular interest – although unclear how relevant to the human disease – they find PNET/gliomas forming in the mice that harbor the GT allele together with p53. Their observations suggest the role of REST as a tumour suppressor - but related to genetic instability, rather than its function as a repressor of neuronal differentiation.

In Figure 5 it looks like cell death is mainly within the SVZ. It might be worth further defining whether the apoptotic cells are TBR2 basal progenitors or radial glia. So a caspase co-stain with TBR2 and RC2 would distinguish if it were apical progenitors/radial glia that were specifically dying. Higher magnification views of TuJ1, MAP2, RC2, TBR2 and caspase would be useful to show. It would be good to clarify this. Do the progenitors that exit cycle turn on MAP2 prematurely and then die? Of is there some 'proper' progression to neuronal differentiation and then cell death? So morphology and location of the dying cells need better characterization.

There is interest in the hypothesis that neuronal differentiation commitment is linked to extension of G1 times. Would it be useful to discuss their findings in this context? Is this transition something that REST would be involved in or monitoring?

The above are my only minor criticisms. This is an impressive study: rigorous, high quality data, novel insights, beautifully presented and of broad interest. I would recommend publication immediately.

---

## [Author Response]

[…]You may want to include some discussion about the relevance of your findings to other tumour types and speculate about inducible REST deletion in adult brain. Please include the quantitation requested in Figure 3 and please consider including some additional markers in Figure 5.

In response to the thoughtful reviews, which were overall quite positive, we did more experiments and rewrote the manuscript to take into account the recommendations. The revised manuscript includes the requested re-quantification and clarification for Figure 3 and the addition of the cell specific markers for Figure 5. To evaluate more thoroughly the relationship between loss of REST, DNA damage, cell cycle and apoptosis, we analyzed our BrdU experiments in different cell populations in addition to what we scored previously, and added a histogram that provides more support for the idea that DNA damage due to premature loss of REST occurs during S Phase in cycling progenitors. That new data is in shown in revised Figure 6. The new experiments prompted a re-working of the Discussion.

Reviewer #1 (Minor comments):

*Overall, the manuscript is well written except a few minor points to be clarified during revision.1) The authors performed REST ChIP on a few gene targets (Figure 2). Where are the REST binding sites, on promoters or RE-1 elements of* Glrp *or* Snap25*? In addition, what are the expression levels of* Glrp *or* Snap25 *in control and REST KO brains?*

The REST binding sites are on consensus RE1 elements. They are located at +275 bp with respect to the transcriptional start site (TSS) in the *Glra* gene, in the first exon, and at +867 bp from the TSS, in the first intron of the *Snap25* gene. This information is now included in the third paragraph of the subsection “Conditional REST gene deficiency in neural progenitors” and the legend to Figure 2.

2) Can the authors clarify CTIP2+ cells (Figure 3) between control and REST KO brains? It appears that CTIP2+ cells are increased (Figure 3), but there are not significant changes reported (Figure 3).

We have revised Figure 3. We apologize for the image shown in old Figure 3. The quantification of CTIP2 was specifically for layer 5 and we did not make that clear. These cells are highly immuno-positive for CTIP2 compared to the cells in layer 6 that are positive for both CTIP2 and TBR1, but at lower levels of immuno-positivity of CTIP2. Additionally, that particular image had a high background that obscured the differences between the high and lower expressing CTIP2+ cells. Therefore, we have replaced the image with a higher resolution image in the revised figure and relabeled the histogram CTIP2high+ and CTIP2low+. In response to this concern, we also went back and re-quantified more sections and the conclusion of no significant differences in the numbers of CTIP2+ cells in layer 5 between knockout and control held.

3) What are the percentage of clCasp3/BrdU + or 53BP1/BrdU + cells in control mice (Figure 6)?

With respect to clCasp3/BrdU + cells in controls, there are too few clCasp3+ cells in controls to make this a meaningful comparison to KO animal and we now reiterate this clearly, with numbers, at the end of the subsection “Loss of REST function alone is responsible for DNA damage”. With respect to 53BP1+/BrdU+ cells, we have added control values in Figure 6 that indicate the same overall percentages of BrdU+ cells in 53BP1+ foci population in control and mutant cells, while the numbers of cells with foci are significantly higher in mutant. We have also added a new histogram showing that in the mutant, the percentage of cells with 53BP1+ foci are much higher in BrdU+ VZ cells compared to BrdU- cells in VZ or CP (Figure 6).

4) Recently, a few non-coding RNAs or splicing variants have been identified in REST locus from many human cancer cells. Therefore, it is possible that REST locus encodes both REST protein and ncRNAs with distinct biological functions. If the authors can determine the expression of non-coding RNAs or splicing variants in both REST KO strategies, it may help the understanding of the biological roles of REST and the non-coding RNAs in REST locus.

Good question. As far as we can tell, in the rodent, exon 2 skipping that generates non coding RNAs in the Rest locus is infrequent and variable in rodents (Chen and Miller, PLOS one, 2013), and exon 3 skipping is not detected in mouse tissue. Further, while authors of the human study assume that exon 2 skipping generates non-coding RNA, our study demonstrates to the contrary, that, at least in mouse, deletion (skipping) of exon 2 can generate a functional protein (REST^C^). We could not detect expression of Rest 2, 3 and 5 isoforms in embryonic mouse brains. Isoform Rest 4 is largely deleted in both knockout strategies in this work. Further probing of this idea, although interesting, seems outside the scope of this work.

Additional data files and statistical comments:

Can the authors provide transcriptional profiling data comparing control with REST KO brains discussed in the manuscript?

This information is included as [Supplementary-material SD1-data] and [Supplementary-material SD2-data] and full data is available on the GEO site with accession numbers listed in the Methods section.

Reviewer #2:

General assessment and major comments:

In Figure 5 it looks like cell death is mainly within the SVZ. It might be worth further defining whether the apopototic cells are TBR2 basal progenitors or radial glia. So a caspase co-stain with TBR2 and RC2 would distinguish if it were apical progenitors/radial glia that were specifically dying. Higher magnification views of TuJ1, MAP2, RC2, TBR2 and caspase would be useful to show. It would be good to clarify this. Do the progenitors that exit cycle turn on MAP2 prematurely and then die? Of is there some 'proper' progression to neuronal differentiation and then cell death? So morphology and location of the dying cells need better characterization.

Thank you for suggesting this experiment. We immuno-labeled with SOX2 antibody to identify all progenitors, and with TBR2 and ClCasp3 antibodies to label basal progenitors and apoptotic cells, respectively (new Figure 5—figure supplement 1). We found no TBR2+ cells in the VZ/ SVZ that were ClCasp3+ (new Figure 5—figure supplement 1). In a second suggested experiment, we co-stained embryonic sections with markers to MAP2, Ki-67 and ClCasp3. As we described before, while a majority of the ClCasp3+ cells were MAP2+ (Figure 5), they were also Ki67- (new Figure 5—figure supplement 1). Thus, it appears that cells are dying as they begin to differentiate into mature neurons and therefore accumulate at the VZ/SVZ border. Because nearly all the improperly located at the VZ/SVZ MAP2 + cells are also ClCasp3+, we cannot order the events. Time course imaging in individual cells might be able to answer this question, but this will require sophisticated approaches that are outside the scope of this study. These new results are now described in the first paragraph of the subsection “REST-deficient neural progenitors undergo p53-mediated cell death linked temporally to distinct neuronal differentiation programs” and discussed further in the Discussion, subsection “The S phase of the cell cycle in neural progenitors is a critical decision point for proper terminal neuronal differentiation”.

There is interest in the hypothesis that neuronal differentiation commitment is linked to extension of G1 times. Would it be useful to discuss their findings in this context? Is this transition something that REST would be involved in or monitoring?

We were able to measure an extension of G1 in our mutant brain, but the effect was quite small. Given this, and confounds of DNA damage and apoptosis, we are not comfortable commenting on a role for REST in this event.